# Chromatin organizer SATB1 controls the cell identity of CD4+ CD8+ double-positive thymocytes by regulating the activity of super-enhancers

Delong Feng [1,6], Yanhong Chen[2,6], Ranran Dai [1,6], Shasha Bian [2,6], Wei Xue [1,6], Yongchang Zhu[2], Zhaoqiang Li[1], Yiting Yang [1], Yan Zhang [1], Jiarui Zhang [1], Jie Bai [2], Litao Qin [2], Yoshinori Kohwi [3], Weili Shi [2], Terumi Kohwi-Shigematsu [3], Jing Ma[4], Shixiu Liao [2,5] ✉ & Bingtao Hao [1] ✉

CD4+ and CD8+ double-positive (DP) thymocytes play a crucial role in T cell development in the thymus. DP cells rearrange the T cell receptor gene *Tcra* to generate T cell receptors with TCRβ. DP cells differentiate into CD4 or CD8 single-positive (SP) thymocytes, regulatory T cells, or invariant nature kill T cells (iNKT) in response to TCR signaling. Chromatin organizer SATB1 is highly expressed in DP cells and is essential in regulating *Tcra* rearrangement and differentiation of DP cells. Here we explored the mechanism of SATB1 orchestrating gene expression in DP cells. Single-cell RNA sequencing shows that *Satb1* deletion changes the cell identity of DP thymocytes and down-regulates genes specifically and highly expressed in DP cells. Super-enhancers regulate the expressions of DP-specific genes, and our Hi-C data show that SATB1 deficiency in thymocytes reduces super-enhancer activity by specifically decreasing interactions among super-enhancers and between super-enhancers and promoters. Our results reveal that SATB1 plays a critical role in thymocyte development to promote the establishment of DP cell identity by globally regulating super-enhancers of DP cells at the chromatin architectural level.

T lymphocytes, a critical component of adaptive immunity, develop as thymocytes in the thymus[1–3]. Most T cells in humans and mice are αβ T cells expressing the TCR consisting of an α- and β-chain. The primary purpose of αβ T-cell development in the thymus is to generate T lymphocytes with highly diverse T-cell receptors (TCRs) to recognize a wide variety of foreign antigens and avoid responses to self-antigens at the same time[1,4]. During differentiation into mature T cells, thymocytes go through several developmental stages, which are characterized by the cell surface expression of CD4 and CD8 proteins. CD4+ CD8+ double-positive (DP) thymocytes are at the center of αβ T-cell development[2]. DP cells generate the αβTCR on the cell surface to recognize cortical epithelial cells expressing Class I or Class II MHC

plus self-peptides for positive and negative selection[1,4]. TCR signaling induces DP cell differentiation into CD4+ single-positive (SP), CD8+ SP, regulatory T cells (Treg), or invariant natural killer T cells (iNKT)[4].

The primary event in DP cells is *Tcra* rearrangement to generate TCRs with an affinity to self-peptide:MHC complexes, which requires that cells do *Tcra* rearrangement efficiently and sense the TCR signal correctly[3]. Due to the low frequency of positive selection, the *Tcra* gene can do multiple rounds of Vα-Jα rearrangement on both alleles to increase the chance of passing positive selection[5]. DP cells also up-regulate recombinase Rag1 and Rag2 expression for efficient rearrangement[6,7]. Cell lifespan is another critical factor for generating diverse TCRα repertoire and the proper development of DP

thymocytes[8]. DP thymocytes can survive for an average of three to four days to allow Jα rearrangement sequentially from the 5′ end to the 3′ end of the Jα array. Depleting the factors that regulate the survival of DP thymocytes (e.g., ROR, an orphan nuclear factor, and YY1, Yin Yang 1) shortens the lifespan and reduces the usage of the 3′ Jα segments[9–11]. However, it is unknown how DP thymocytes orchestrate expression programs for efficient *Tcra* rearrangement and proper selection.

Special AT-rich binding protein 1 (SATB1) is a chromatin organizer that plays certain roles in many tissues, including skin[12], tooth[13,14], and liver[15], although it is not ubiquitously expressed. Studies also showed that SATB1 is involved in embryogenesis[16,17], neurogenesis[14,18], hematopoiesis[19,20], and erythropoiesis[21,22]. Nevertheless, most of the studies on SATB1 focused on its role in T-cell development, partially due to the highest SATB1 expression during thymocyte development, especially at the DP stage[6,23–26]. *Satb1*-deleted mice displayed a smaller thymus, increased proportion of DP thymocytes, and fewer CD4/CD8 single-positive cells, indicating a blockage at the DP stage during thymocyte development[23]. The development blockage may be at least partially due to insufficient *Tcra* rearrangement and impaired positive selection[6,27]. In post-selection thymocytes, SATB1 also plays a role in activating lineage-specifying genes, including ThPOK, Runx3, CD4, CD8, and Foxp3[25,28]. SATB1-deficient thymocytes display inappropriate T-lineage determination after MHC I- and II-mediated selection, and deficient differentiation of regulatory T cells ($T_{reg}$)[28]. These findings indicate the critical role of SATB1 in DP thymocytes, although the detailed molecular mechanisms remain elusive.

SATB1 can activate or repress gene transcription by recruiting p300/CBP-associated factor (PCAF) or histone deacetylase, respectively[29–31]. SATB1 functions as a pioneer factor in establishing $T_{reg}$ cell-specific super-enhancers, which is crucial for $T_{reg}$ cell lineage specification in the thymus[28]. More studies have been addressing the roles of SATB1 in chromatin organization[6,21,24,25,32–34], which was suggested when it was identified as a nuclear protein that specifically binds to genomic sequences with a highly nucleotide base unpairing property[35,36]. An early report showed that SATB1 induced a unique transcriptionally active chromatin structure at the T helper 2 ($T_H$2) cytokine locus during $T_H$2 cell activation[24]. SATB1 directly controls the regulatory elements of several lineage-specifying genes, including *Zbtb7b* (encoding ThPOK), *Runx3*, *Cd4*, *Cd8*, and *Foxp3*[25]. SATB1 also mediates the DP-specific interaction between the anti-silencer element and *Rag1* gene promoter for the high expression of Rag1 and Rag2 in DP thymocytes[6]. However, if and how the expression of genes in DP thymocytes is regulated at the level of SATB1-mediated chromatin organization remains unknown.

In this study, we analyzed the development of SATB1-deficient thymocytes using the single-cell RNA sequencing (scRNA-seq) technique and found that *Satb1* deletion changes the expression of cell identity genes of DP thymocytes. Further analysis shows that SATB1 is essential for activating super-enhancers by reorganizing their interactions. We also analyzed the regulation and function of two genes encoding transcription factors BCL6 and ETS2, which are regulated by SATB1 and super-enhancers in DP thymocytes. These findings indicate that SATB1 controls the cell identity of double-positive thymocytes by regulating the activity of super-enhancers.

## Results

### *Satb1* deletion changed the cell identity of CD4⁺ CD8⁺ double-positive thymocytes

To investigate the role of SATB1 in thymocyte development, we employed single-cell RNA sequencing on all thymocytes from 6-week-old female mice (*Satb1*^f/f × *vav*-cre mice) in which the *Satb1* gene was deleted in hematopoietic stem cells using *vav*-cre transgene[6,27]. A total of 13948 cells consisting of 6844 SATB1-deficient and 7104 control thymocytes passed the quality control criteria. Cells were separated into sixteen clusters according to gene expression and cell cycling

progress (Fig. 1a and Supplementary Fig. 1a). Our previous study showed that SATB1 regulates recombinase Rag1 expression in DP thymocytes[6]. The scRNA-seq data showed that reduced Rag1 expression occurred on the level of a single cell and the number of cells (Fig. 1b and Supplementary Fig. 1b, c). We observed a decrease in the DN2/3 thymocyte number, an increase in the DN4/ISP/DP cell number, and a reduction of the CD4/CD8 SP cell numbers in the SATB1-deficient thymus (Fig. 1a and Supplementary Fig. 1d), which is consistent with our and previous reported flow cytometry analysis (Supplementary Fig. 1e)[23]. The B-cell number also increased in the SATB1-deficient thymus (Fig. 1a and Supplementary Fig. 1a). A recent report showed that Satb1 plays a role in B-cell survival and maturation[37]. The increased B-cell number may be an intrinsic feature due to the *Satb1* deletion in hematopoietic stem cells. We noticed that most of the DP thymocytes from the SATB1-deficient thymus were enriched in cluster three, while the cells in cluster three were rare in the wild-type (Fig. 1a and Supplementary Data S1), suggesting that cluster three was generated by *Satb1* deletion. The result indicated that SATB1 deficiency changed the cell identity of DP thymocytes.

The scRNA-seq analysis showed 700 differentially expressed genes (DEGs) in DP thymocytes with 383 upregulated and 317 downregulated genes (Supplementary Data S1). *Satb1* deletion reduced expression of some DP feature genes, including recombinase genes *Rag1* and *Rag2*, cell surface marker gene *Cd8a*, and chemokine receptor gene *Ccr9*. However, scRNA-seq has a limitation in the detection of low expressed genes due to its low efficiency[38]. To avoid the limitation of scRNA-seq data, we analyzed the gene expression profile of SATB1-deficient DP thymocytes with independent bulk RNA-seq data (Fig. 1c and Supplementary Fig. 2a). There were 928 downregulated genes and 576 upregulated genes in SATB1-deficient DP thymocytes (Supplementary Fig. 2b, c and Supplementary Data S1). DEGs from the scRNA-seq is highly correlated to that of the bulk RNA-seq ($r = 0.79$, Supplementary Fig. 2d).

We noticed that some of the upregulated genes in SATB1-deficient DP cells are the feature genes in the early stages of thymocyte development, such as the *Il2ra* gene encoding CD25, a cell surface maker of DN2/3 cells (Fig. 1c and Supplementary Fig. 1c). We also confirmed some upregulated genes by quantitative PCR (Supplementary Fig. 2e). To learn expression characteristics of the upregulated genes during thymocyte development, we analyzed expression data of nine developmental stages (from DN1 to SP) obtained from ImmGen Datasets[39]. Most of the upregulated genes are highly expressed in thymocyte development earlier stages like DN1 and DN2a (Fig. 1d, e and Supplementary Fig. 2f). About 27% of the upregulated genes have an expression peak in the DN1 stage, while 10% of these genes have a peak in DP cells (Supplementary Fig. 2f). In total, 77% of upregulated genes have an expression peak in earlier stages (from DN1 to ISP). To confirm SATB1 repressing the expression of the genes expressed explicitly in earlier stages, we did the Gene Set Enrichment Analysis (GSEA) with the DN1 and DN3-specific genesets generated from ImmGen Datasets (Supplementary Data S2). The data showed that the DN1 and DN3 gene sets were significantly repressed by SATB1 (Fig. 1f). Many downregulated genes in SATB1-deficient DP thymocytes were highly expressed in DP and SP thymocytes (Fig. 1d, e). In all, 36% of downregulated genes have an expression peak in the DP stage (Supplementary Fig. 2f). The Gene Ontology enrichment analysis also showed that the downregulated genes are involved in T-cell activation, a critical feature of DP thymocytes (Supplementary Fig. 2g and Supplementary Data S3). These data suggested that SATB1 controls the cell identity of DP thymocytes by activating the DP-specific genes and repressing the genes expressed in earlier stages.

### SATB1 binds and activates super-enhancers of DP thymocytes

It has been shown that super-enhancers drive expression of genes that define cell identity[40]. The previous study showed that SATB1 plays a

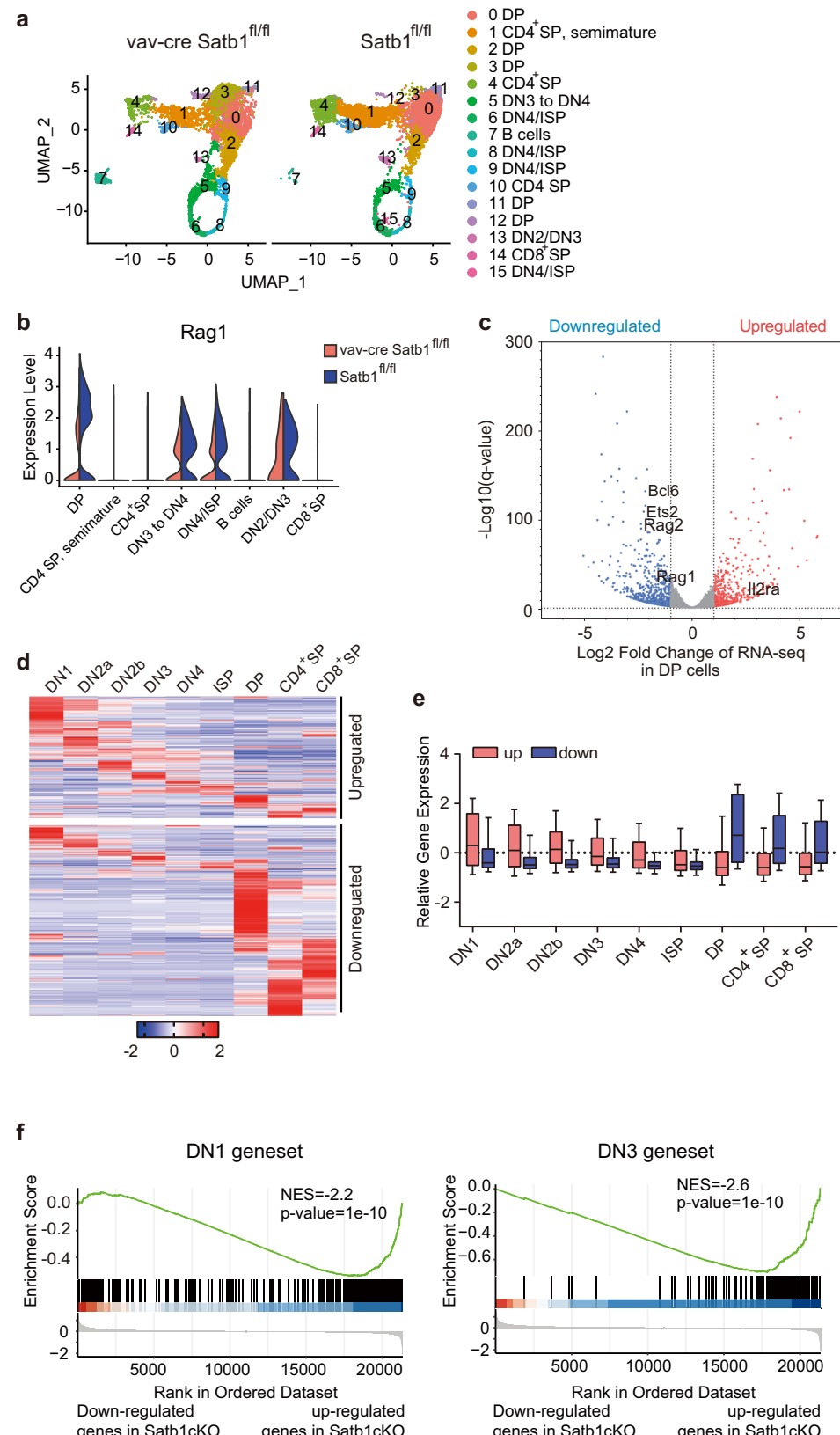

role in activating super-enhancers in Foxp3+ regulatory T cells[28]. To explore the role of SATB1 in organizing super-enhancers of DP thymocytes, we identified 246 super-enhancers of DP thymocytes using the algorithm ROSE[40] with the published ChIP-seq data of histone H3 acetylated at Lys27 (H3K27ac) in sorted DP thymocytes[28] (Fig. 2a and Supplementary Data S4). It was reported that the super-enhancer

landscape is established a priori to the stage in which the gene expression is controlled[41]. We analyzed chromatin accessibility of DP super-enhancer regions in each stage of thymocyte development using ATAC-seq data from ImmGen Datasets[39]. Accessibility of super-enhancer regions increased from DN1, peaked at ISP, and dropped in DP (Supplementary Fig. 3a), indicating the establishment of DP super-

**Fig. 1 | SATB1 controls cell identity of DP thymocytes by repressing early expressing genes and activating DP-specific genes. a** Two-dimensional UMAP plot of thymocytes single-cell transcriptomes in *vav*-cre × *Satb1*^fl/fl (Satb1cKO) and *Satb1*^fl/fl (Satb1WT) mice. Each dot represents one cell and different areas identified by unsupervised clustering. **b** Violin plots showing *Rag1* gene expression per cell in difference cell subsets from Satb1cKO and Satb1WT mice. **c** Volcano plot showing bulk RNA-seq differentially expressed genes (Satb1cKO vs Satb1WT) in sorted DP thymocytes. Three biological replicates were performed. The *x*- and *y* axis represent log2(fold change) and −log10(*P* value), respectively. Some genes are annotated. **d** Heatmap and **e** boxplot showing relative gene expression of the differentially expressed genes during the differentiation from DN1 to SP thymocytes. The differentially expressed genes contain 893 downregulated and 556 upregulated genes Box plots show median (center line), interquartile range (box), and tenth and ninetieth percentiles (whiskers). DEGs are from bulk RNA-seq data in **c**. Gene expression data from DN1 to SP stages are derived RNA-seq data from ImmGen datasets (GSE109125). **f** Gene-set enrichment analysis (GSEA)-enrichment plots for DN1- or DN3-specific genes showing a significant enrichment of the geneset in upregulated genes of Satb1cKO DP thymocytes. DN1 and DN3 genesets were generated according to the highest expression stage from DN1 to SP stages. The expression data are from bluk RNA-seq of sorted DP thymocytes from WT and Satb1cKO mice. Nominal *P* value, calculated as two-sided *t* test, no adjustment since only one geneset was tested. Source data are provided as a Source Data file.

enhancers in the ISP stage. Many super-enhancers are associated with known DP signature genes, such as *Tcra*, *Rag1*, *Cd8a*, *Cd4*, etc. (Fig. 2a). Expression of the DP super-enhancer-associated genes increased gradually during development and peaked at the DP stage, and dropped in the SP stage (Fig. 2b and Supplementary Fig. 3b), suggesting that super-enhancers control the expression of cell identity genes in DP thymocytes.

To learn the relationship between SATB1 and super-enhancers, we reanalyzed the previous SATB1 ChIP-seq data with sorted DP thymocytes[28]. It was reported that active promoters can produce false-positive peaks in ChIP-seq experiments[42]. Therefore, we performed SATB1 ChIP-seq in Satb1-deficient thymocytes from Satb1^f/f × *CD4*-cre mice. The results showed that the SATB1 binding in active regions was specific in SATB1-expressing WT cells (Fig. 2c). We noticed that many super-enhancers had high SATB1 occupancy, like the *Cd4* locus (Fig. 2c). Then we did an overlapping analysis of SATB1 peaks with published ChIP-seq data of the active histone-modification markers histone H3K4 monomethylation (H3K4me1)[28], H3K4me3[43], H3K27ac[28], and chromatin organization complex Rad21, Nipbl[44], and CTCF[45]. SATB1-binding sites have high RAD21, CTCF, and Nipbl occupancy, especially for clusters 1 and 3 (Supplementary Fig. 3c, d). SATB1 occupancy overlapped with H3K4me3 in many sites and displayed the same direction in the average plot (Supplementary Fig. 3e, f). While SATB1 binding displayed an opposite direction with H3K4me1 and H3K27ac on average (Supplementary Fig. 3e). We noticed that SATB1 occupancy had the same direction in cluster 3, while cluster 4 displayed a strong opposite direction between SATB1 and H3K27ac (Supplementary Fig. 3f). It was reported that SATB1 had either transcriptional repression or activation, depending on the histone acetylation status by recruiting the histone deacetylase either HDAC1 or acetylase CBP/p300[31], respectively, which would explain the two H3K27ac modification status on SATB1-occupied sites.

The SATB1 signals were enriched in super-enhancers (Fig. 2d and Supplementary Fig. S3g, h). About 95% of super-enhancers have SATB1 occupancy, while it is only 20% for traditional enhancers (Supplementary Fig. 3h). We observed a correlation between SATB1 occupancy and H3K27ac modification in super-enhancer regions (Fig. 2d), indicating that SATB1 may recruit histone acetylases instead of deacetylases to super-enhancers. Consistent with it, SATB1 deficiency leads to a significant decrease in H3K27 acetylation in super-enhancer regions (Fig. 2e). The average intensity of H3K4me1 in super-enhancers was reduced in SATB1-deficient cells (Supplementary Fig. 3j). SATB1 deficiency changed the landscape of super-enhancers in DP thymocytes, 118 super-enhancers were lost (lost SEs), and 106 new super-enhancers gained (gained SEs) (Fig. 2f). The result showed that SATB1 regulated activity of super-enhancers in DP thymocytes.

Then we analyzed the expression of SE-associated genes in SATB1-deficient cells using the bulk RNA-seq data. The GSEA analysis showed that all SE-associated and lost-SE-associated genes are significantly enriched in downregulated genes of SATB1-deficient cells (Fig. 2g). In all, 67% of SE-associated genes were downregulated and 33% upregulated (Fig. 2h). We observed a significant downregulation of lost-SE-associated genes in SATB1-deficient DP thymocytes and a slightly reduced expression of the maintained-SE-associated genes (Fig. 2i). SE-associated genes are more sensitive to SATB1 deficiency than traditional enhancer-associated genes (Fig. 2j). On the contrary, the gained-SE-associated genes were enriched in the upregulated genes of SATB1-deficient cells (Fig. 2i and Supplementary Fig. S3k). The gained SEs were generally transferred from traditional enhancers, and most gained-SE-associated genes have an expression peak in the DP stage (Supplementary Fig. 3l, m), indicating that these genes did not represent the earlier stage genes repressed by SATB1. Taken from all the above results, it was suggested that SATB1 regulated the DP signature genes by activating super-enhancers.

## SATB1 clusters are associated with cell identity genes

The SATB1-binding sites were clustered in the genome and displayed a super-enhancer-like distribution (Fig. 2c). We did the ROSE analysis with SATB1 ChIP-seq data and identified 743 SATB1 super-clusters (SCs) (Fig. 3a and Supplementary Data S5). The SATB1-SC regions associated genes consisted of many DP signature genes, such as *Tcra*, *Rag2*, *Ets1*, *Cd8a*, *Cd4*, etc. About 64% of super-enhancers were overlapped with SATB1 super-clusters (Fig. 3b). SATB1-SC regions had a higher level of active histone modification H3K27ac than non-clustering SATB1-binding regions (Fig. 3c). The H3K27 acetylation level of the SATB1-SC regions reduced in SATB1-deficient cells, while non-clustering SATB1-binding sites displayed increased H3K27ac (Fig. 3c). *Satb1* deletion downregulated most of the SATB1-SC-associated genes (Fig. 3d), suggesting that SATB1 clustering tended to activate the expression of their associated genes. We analyzed the expression of the SATB1-SC-associated genes during thymocyte development using the gene expression data from ImmGen Datasets. SATB1-SC-associated genes are highly expressed in DP thymocytes, and many of them are specifically expressed in the DP stage (Fig. 3e and Supplementary Data S6). The Gene Ontology (GO) term enrichment analysis showed that these genes enriched in DP associating functions, such as T-cell activation and V(D)J recombination (Fig. 3e). These results indicated that SATB1 clustering promotes histone acetylation and actives the DP signature genes.

## SATB1 mediated the chromatin interactions of super-enhancers

It was reported that chromatin organization changed during thymocyte development[41]. We analyzed chromatin interaction changes in the SE and TE regions from DN1 to DP stages using the published Hi-C[41]. Interactions in the SE regions displayed two obvious increased transitions from DN2 to DN3 and DN4 to DP, respectively, while transitions in TE regions are relatively weak from DN4 to DP (Supplementary Fig. 4a). The SATB1-binding clusters also displayed a similar feature. The previous report showed that SATB1 regulated the expression of V(D)J recombinase genes *Rag1* and *Rag2* in DP thymocytes by mediating an enhancer–promoter interaction[6]. To learn the role of SATB1 in chromatin organization of SE regions in DP thymocytes, we performed Hi-C experiments with sorted DP thymocytes from *Satb1*^f/f and *Satb1*^f/f ×*vav*-cre mice. Pairwise Pearson correlation analysis of the top

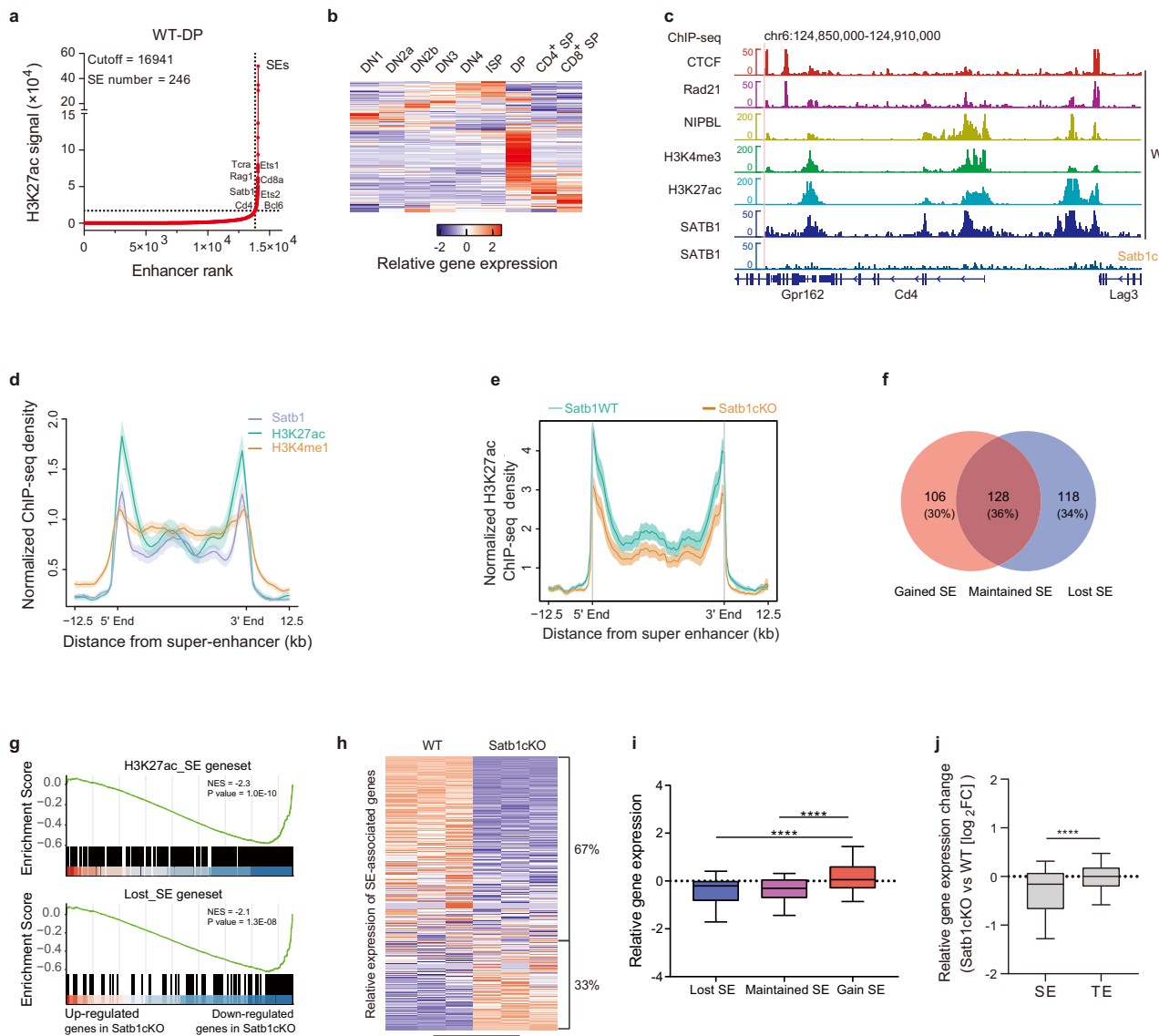

**Fig. 2 | SATB1 binds and activates super-enhancers of DP thymocytes. a** Hockey-stick plots showing super-enhancers based on normalized H3K27ac signals and rank in Satb1WT DP thymocytes using the ROSE algorithm. Some super-enhancer-associated genes are highlighted. The H3K27ac ChIP-seq data are from dataset DRP003376[28]. **b** Heatmap showing relative expression of super-enhancer-associated genes during the differentiation from DN1 to SP thymocytes. Gene expression data are derived RNA-seq data from ImmGen datasets (GSE109125). **c** Normalized H3K27ac, H3K4me3, CTCF, Rad21, Nipbl, and SATB1 ChIP-seq profiles at the *Cd4* locus in WT DP cells and the SATB1 ChIP-seq in SATB1-deficient thymocytes (*Satb1*[f/f] × *CD4*-cre). H3K27ac and H3K4me1 ChIP-seq data are from DRP003376[28]; Satb1 ChIP-seq in WT DP from GSE90635[25]; H3K4me3 ChIP-seq, GSE21207[43]; Rad21 and Nipbl ChIP-seq, GSE48763[44]; CTCF ChIP-seq, GSE141223[45]. **d** Line plot displaying H3K27ac, H3K4me1, and SATB1 ChIP-seq signal density at WT DP super-enhancer regions. Data represent the mean ± SEM ChIP-seq data are same as (**c**). **e** Line plot displaying H3K27ac signal density in Satb1WT vs Satb1cKO at WT DP super-enhancer regions. Data represent the mean ± SEM. **f** Venn diagram showing super-enhancer changes in DP thymocytes of Satb1cKO mice. H3K27ac ChIP-seq signals in Satb1cKO DP thymocytes were used for SE calling. Gained SE,

new super-enhancers in Stab1cKO DP thymocytes; maintained SE, maintained super-enhancers in Satb1cKO; lost SE, not super-enhancers in Satb1cKO. **g** GSEA-enrichment plots of the differentially expressed genes enriched for the WT DP super-enhancer associated geneset or the Satb1cKO DP lost super-enhancer associated geneset. Details of genesets can be found in Supplemental Data S4. Bulk RNA-seq data of WT and Satb1cKO DP thymocytes were used here. Nominal *P* value, calculated as two-sided *t* test, no adjustment since only one geneset was tested. **h** Heatmap showing the expression of WT DP super-enhancer-associated genes in WT and Satb1cKO DP thymocytes. Details of data can be found in Supplemental Data S6. **i** Relative expression of genes associated with super-enhancers gained (112 genes), maintained (107 genes), or lost (107 genes) in Satb1 KO DP cells. Gained vs Maintained, *P* = 1.66e-06; Gained vs Lost, *P* = 2.66e-06. Box plots show median (center line), interquartile range (box) and 10th and 90th percentiles (whiskers). *P* value <0.05, ***P* value <0.001 by Kruskal−Wallis test, followed by Dunn's multiple comparisons test. **j** Relative expression of the super-enhancer (SE) or traditional enhancer (TE) associated genes in Satb1cKO DP cells. (SE, 223; TE, 11905; *P* = 4.99e-14). ****P* value <0.0001 by two-sided Student's *t* test. Source data are provided as a Source Data file.

---

1000 variable bins showed that biological replicates were reproducible (Supplementary Fig. 4b). Visualization of Hi-C data revealed minor alterations of chromatin organization in SATB1-deficient DP thymocytes (Supplementary Fig. 4c, d). *Satb1* deletion induced a modest increase in diagonal interactions in most chromosomes. The contact

density decaying curves showed that the interactions less than 2 Mb in length slightly increased in SATB1-deficient thymocytes (Supplementary Fig. 4e). *Satb1* deletion has few effects on compartments (Supplementary Fig. 5a). *Satb1* deletion did not change the number of TADs but slightly reduced sizes of TADs, especially the TADs containing SEs

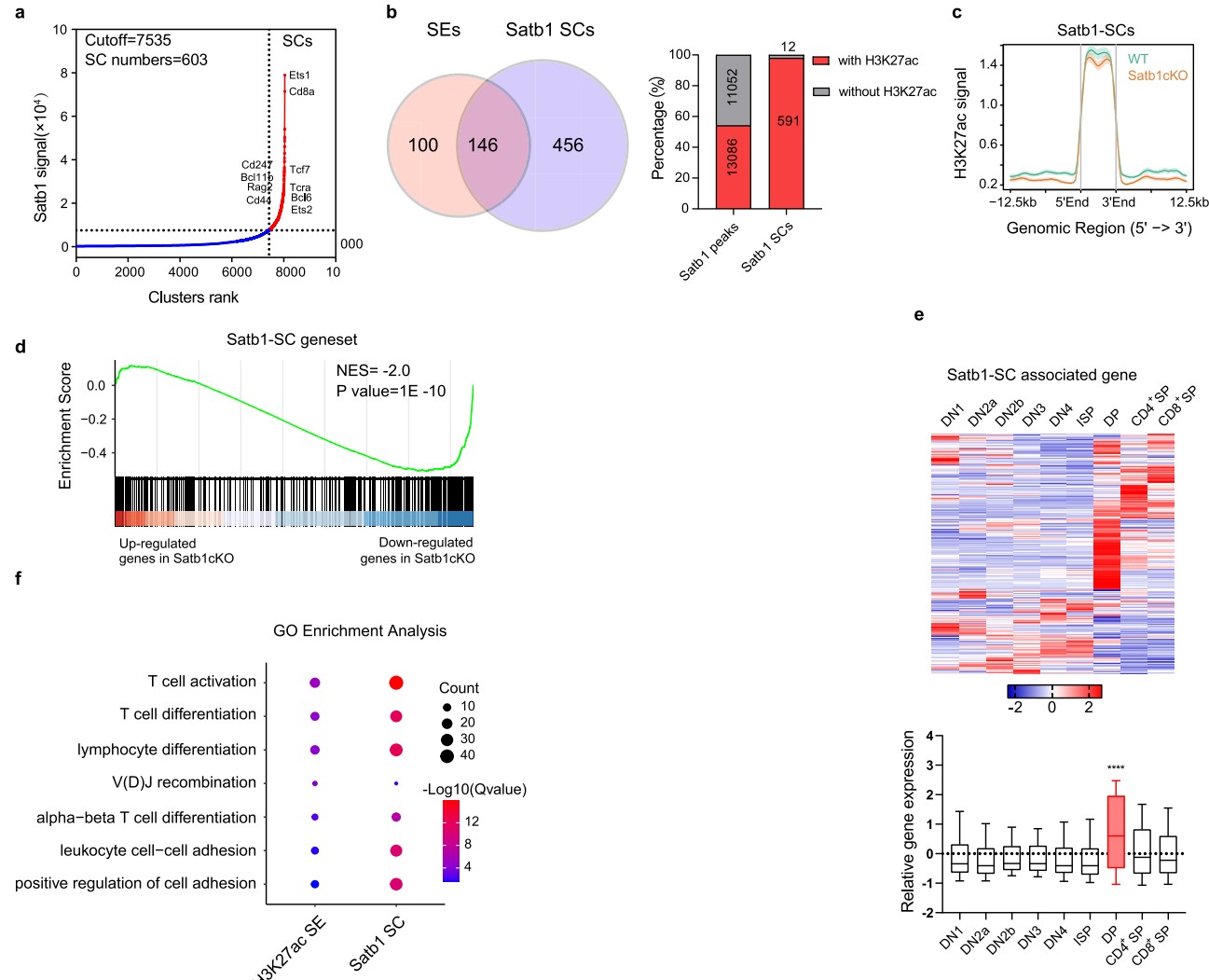

**Fig. 3 | SATB1 clusters are associated with cell identity genes. a** Hockey-stick plots showing clusters ranked by normalized Satb1 ChIP-seq signals in DP thymocytes using the ROSE algorithm. SCs, super-clusters. Details of Satb1-super-cluster and their associated genes can be found in Supplemental Data S5. **b** Venn diagram (left) showing the overlap of Satb1 super-clusters and H3K27ac super-enhancers in WT DP thymocytes. Column plot (right) showing the overlap of Satb1-binding sites with H3K27ac peaks. ChIP-seq data are the same as Fig. 2c. **c** Line plot showing H3K27ac signals of Satb1 super-cluster regions in WT and Satb1cKO DP thymocytes. Data represent the mean ± SEM. **d** GSEA-enrichment plot for Satb1 super-cluster associated genes showing a significant enrichment of the geneset in downregulated genes of Satb1cKO DP thymocytes. Nominal *P* value, calculated as two-sided *t* test,

no adjustment since only one geneset was tested. **e** Heatmap (up) and boxplot (bottom) showing relative gene expressions of SATB1 super-culster-associated genes (795 genes) during the differentiation from DN1 to SP thymocytes. *P* value between DP and other stages as following; DN1, 6.50e-30; DN2a, 2.92e-41; DN2b, 5.23e-24; DN3, 1.32e-26; DN4, 1.08e-38; ISP, 4.00e-42; CD4SP, 2.97e-15; CD8SP, 1.85e-19. Details of data can be found in Supplementary Data S6. ****P value <0.0001 by Kruskal–Wallis test, followed by Dunn's multiple comparisons test. **f** Gene ontology analysis on the genes associated with super-enhancers by enrichment for H3K27ac or super-clusters of Satb1. The top seven biological processes ranked by significance (−log(q-value)) are depicted. Source data are provided as a Source Data file.

(Supplementary Fig. 5b). The TAD strength almost kept the same level in SATB1-deficient cells (Supplementary Fig. 5c). These Hi-C results for *Satb1*^f/f × *vav*-cre mice are similar to those for *Satb1*^f/f × *CD4*-cre mice[36]. Taken together with the Hi-C results, our chromatin interaction studies suggest that the role of SATB1 in the organization of chromatin is not genome-wide, but it is restricted to specific regions, i.e., super-enhancers.

To characterize chromatin interactions affected by *Satb1* deletion, we generated a chromatin interaction matrix with a 50 kb resolution and identified significantly differential interactions using the Bioconductor package multiHiCcompare[46]. We compared 766,111 chromatin interactions and found 500 significantly increased interactions and 411 decreased interactions. Most of the distances between two anchors of decreased interactions are around 10 kb, less than increased interactions (Fig. 4a). In addition, there are high SATB1,

H3K27ac, and CTCF ChIP-seq signals in the anchor regions of decreased interactions (Fig. 4b and Supplementary Fig. S5d). These data showed that, unlike CTCF and Cohesin, which mediate long-range chromatin interactions, SATB1 mainly mediates chromatin relatively short-range chromatin interactions (from tens to 100 kb).

To identify functional interactions among regulatory elements and their target promoters, we explored the chromatin loop structure using the program Fit-Hi-C(V2), a tool for assigning statistical confidence estimates to intra-chromosomal contact maps[47]. We obtained 241,088 and 103,887 loop-like contacts from two WT Hi-C data, 289,023 and 144,941 contacts from two *Satb1*-cKO Hi-C data. Pile-up analysis showed that the average strength of all chromatin loops displayed a mild reduction in SATB1-deficient DP thymocytes (Fig. 4c and Supplementary Fig. S5e). The reduction was also observed in the loops associated promoters, super-enhancers, and

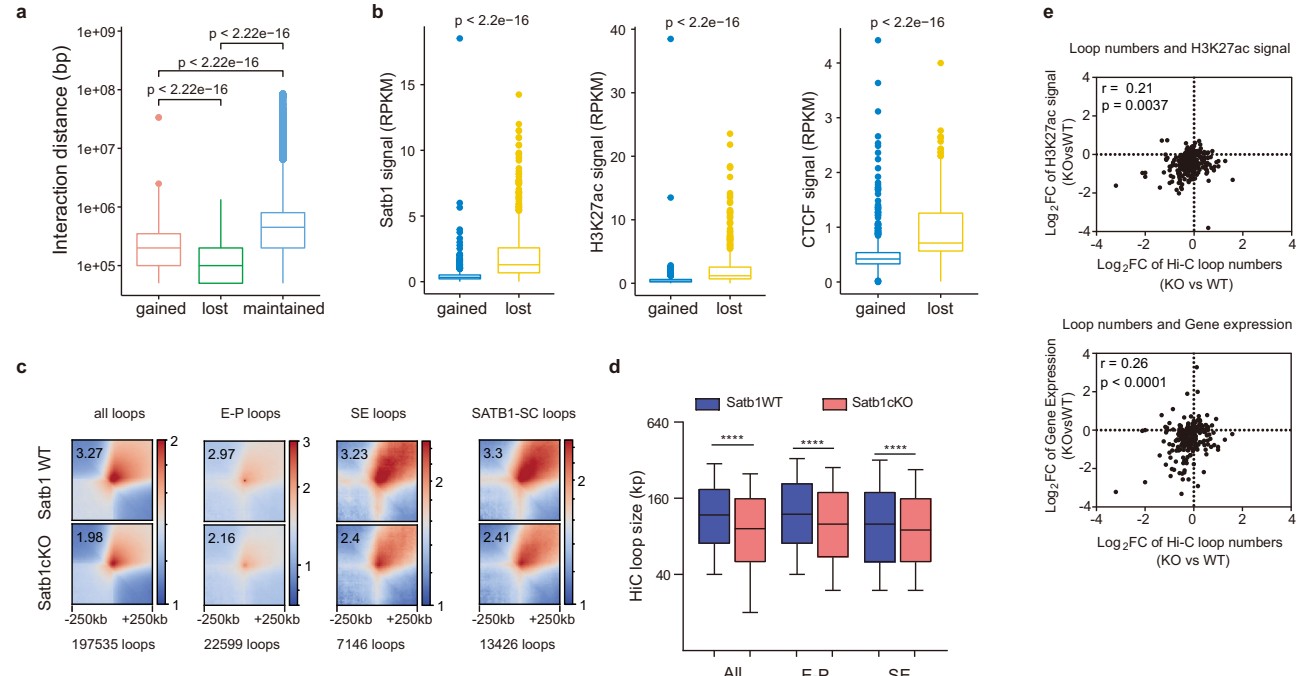

**Fig. 4 | SATB1 mediated the chromatin interactions of super-enhancers.**
**a** Boxplot showing the contact distance changes between two anchors of sig-nificantly changed contacts (50 kb bin) in Satb1-deficient DP thymocytes. Gained, significantly increased interactions; lost, significantly decreased interactions; maintained, no significant changed interactions. The interactions with 0 bp dis-tance were deleted. (gained, $n = 500$, $P = 2.2e\text{-}16$; lost, $n = 411$, $P = 2.22e\text{-}16$; main-tained, $n = 765200$, $P = 2.22e\text{-}16$). Box plots show median (center line), interquartile range (box), and fifth and 95th percentiles (whiskers). The one-tailed Wilcoxon signed-rank test was used. **b** Boxplot of ChIP-seq signals in anchors of differential chromatin interactions (50 kb bin). 648 gained SEs and 549 lost SEs, $P = 2.22e\text{-}16$. Box plots show median (center line), interquartile range (box), and fifth and 95th percentiles (whiskers). The one-tailed Wilcoxon signed-rank test was used.
**c** Aggregate interactions of all loops, enhancer–promoter (E-P) loops, super-

enhancer (SE) loops, and Satb1-SC loops identified from Hi-C data of Satb1WT thymocytes. Pile-up plots showing contacts of ± 250 kb regions around loops in WT or Satb1cKO Hi-C (10 kb resolution). The numbers in pile-up plots represent the value of the highest contacts in the heatmap. SE loops, loops with one or two anchors in WT-SE regions. **d** Boxplot of loop sizes of all loops (WT, 392,910 loops; KO, 386,029 loops, $P = 0$), E-P loops (WT, 34,119 loops; KO, 31,808 loops; $P = 3.42e\text{-}165$), SE loops (WT, 13,507 loops; KO, 12,758 loops; $P = 5.40e\text{-}16$) identified from Hi-C data of Satb1WT thymocytes. Box plots show median (center line), interquartile range (box) and 10th and 90th percentiles (whiskers). ****$P$ value <0.0001 by two-sided Student's $t$ test. **e** Pearson's correlation analysis of super-enhancer associated H3K27ac signals (up, $P = 0.0037$) or relative gene expression (down, $P = 3.94e\text{-}05$) with numbers of loops associated with WT super-enhancers. $P$ values by two-sided F test. Source data are provided as a Source Data file.

Satb1 clusters. We noticed that loop sizes were reduced in SATB1-deficient cells with a median loop size of ~120 kb in WT and ~100 kb in *Satb1*-cKO cells, which was also observed in E-P loops and SE-associated loops (Fig. 4d). Super-enhancers are clusters of enhancers occupied by a high density of transcription factors, co-factors, chromatin regulators, and RNA polymerase II[48]. According to the phase separation model of super-enhancers, SEs may exist in two states in nuclei, one is a highly active condensate, and the other is a random loose state[48]. Reduced sizes of SE-associated loops may be caused by more loose states of super-enhancers in nuclei after Satb1 deletion, which keeps short-distance chromatin interactions but loses long-distance interactions. It suggested that SATB1 may be involved in stabilizing interactions between regulatory elements and compacting chromatin condensates.

To explore the relationship between SE-associated loops and SE activity, we analyzed loop numbers and loop average contacts of each super-enhancer and did correlation analysis with histone-modification H3K27 acetylation of SE regions and expression of SE-associated genes. 68% of super-enhancers have reduced loop num-bers in Satb1-deficient cells (Fig. 4e) and about 80% of super-enhancers had reduced loop densities (average loop contacts) (Supplementary Fig. 5g). The numbers and densities of SE-associated loops is highly correlative, except for 10% of SEs where loop numbers increased but density decreased (Supplementary Fig. 5g). Both the numbers and density of SE-associated loops were correlated with H3K27ac and gene expression (Fig. 4e and Supplementary Fig. S5h–j).

These data indicated that SATB1 promotes internal interactions of SEs and interactions between SEs and promoters to regulate gene expression by forming compacted chromatin organization. This agrees with previously reported SATB1 functions in establishing densely looped chromatin structure in many loci, including the cytokine gene locus[24] and the Rag locus[6].

### SATB1-regulated transcription factors Bcl6 and Ets2 by mediating chromatin topology
SATB1 regulates the DP signature genes, including *Tcra*, *Rag1*, *Rag2*, *Cd4*, and *Ets1*, most of which have been proved to play an essential role in DP thymocytes. We also observed that some genes, like *Bcl6* and *Ets2*, were regulated by SATB1 and super-enhancers in DP thymocytes (Fig. 1c). We further explored the regulation of the *Bcl6* and *Ets2* genes in DP thymocytes. B-cell lymphoma 6 protein (Bcl6) is a zinc finger transcription repressor and is found to be frequently translocated in diffuse large B-cell lymohoma[49–52]. It is a master transcription factor for the differentiation of Follicular Helper T cells (Tfh)[53]. ETS proto-oncogene 2 (Ets2) belongs to the ETS family of transcription factors and is involved in stem cell development, cell senescence and death, tumorigenesis, and thymocyte development[54–58]. These two genes were reported involved in thymocyte development, but their regula-tions remain elusive[56,59].

We observed that SATB1 occupied the loci of the *Bcl6* and *Ets2* genes in DP thymocytes (Fig. 5a, b). There are super-enhancers located at the *Bcl6* upstream and *Ets2* downstream, respectively (Fig. 5a, b).

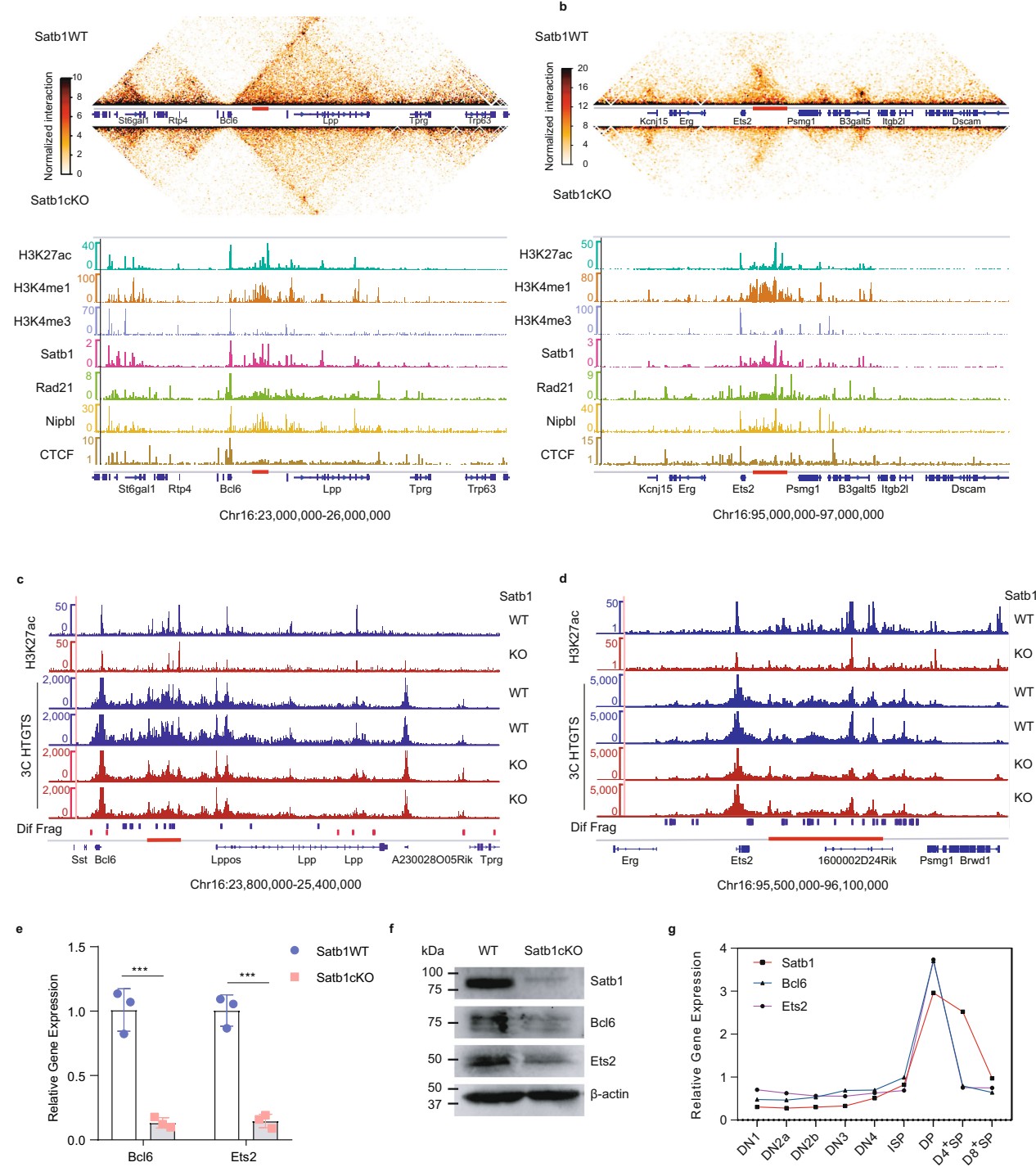

**Fig. 5 | SATB1 regulates transcription factors Bcl6 and Ets2 by mediating promoter–enhancer interactions. a, b** In situ Hi-C chromatin interaction heatmaps (up) of Satb1WT and Satb1cKO DP thymocytes for the Bcl6 (**a**) and Ets2 locus (**b**). H3K27ac, H3K4me1, H3K4me3, Satb1, Rad21, Nipbl, and CTCF ChIP-seq tracks (bottom) of Satb1WT cells for each locus. ChIP-seq data are same as Fig. 2c. **c, d** ChIP-seq and 3C-HTGTS at the *Bcl6* and *Ets2* loci. The 3C-HTGTS bait is at the *Bcl6* or *Ets2* promoters. **e** Relative expression of Bcl6 and Est2 in Satb1WT and Satb1cKO DP thymocytes detected by reverse-transcribed quantitative PCR. The

expressions are normalized to the *Actb* gene and then to WT. $n = 3$ for each group, *Bcl6*, $P = 0.0009$; *Ets2*, $P = 0.0004$. The data represent mean ± SD of three experiments. ***$P$ value <0.001 by two-sided Student's $t$ test. **f** Western blot showing protein expression of Satb1, Bcl6, and Ets2 in WT and Satb1cKO thymocytes. β-actin was used as an internal control. **g** The expression profiles of *Satb1*, *Bcl6*, and *Ets2* during the differentiation of DN1 into DP thymocytes. Source data are provided as a Source Data file.

These two super-enhancers span more than 100 kb and were characterized of H3K27 acetylation, Satb1, cohesin, and CTCF binding (Fig. 5a, b). More important, the super-enhancers and *Bcl6* or *Ets2* are in a topology associating domain (TAD) or sub-TAD. The Hi-C data showed that the chromatin interactions in the two loci, including

super-enhancers and promoters, decreased dramatically in SATB1-deficient DP thymocytes (Fig. 5a, b and Supplementary Fig. S6a–c).

To confirm enhancer–promoter interactions of the two genes, we performed 3C-HTGTS assay[60], a new technique like the Circular Chromosome Conformation Capture (4C). We prepared a 3C library

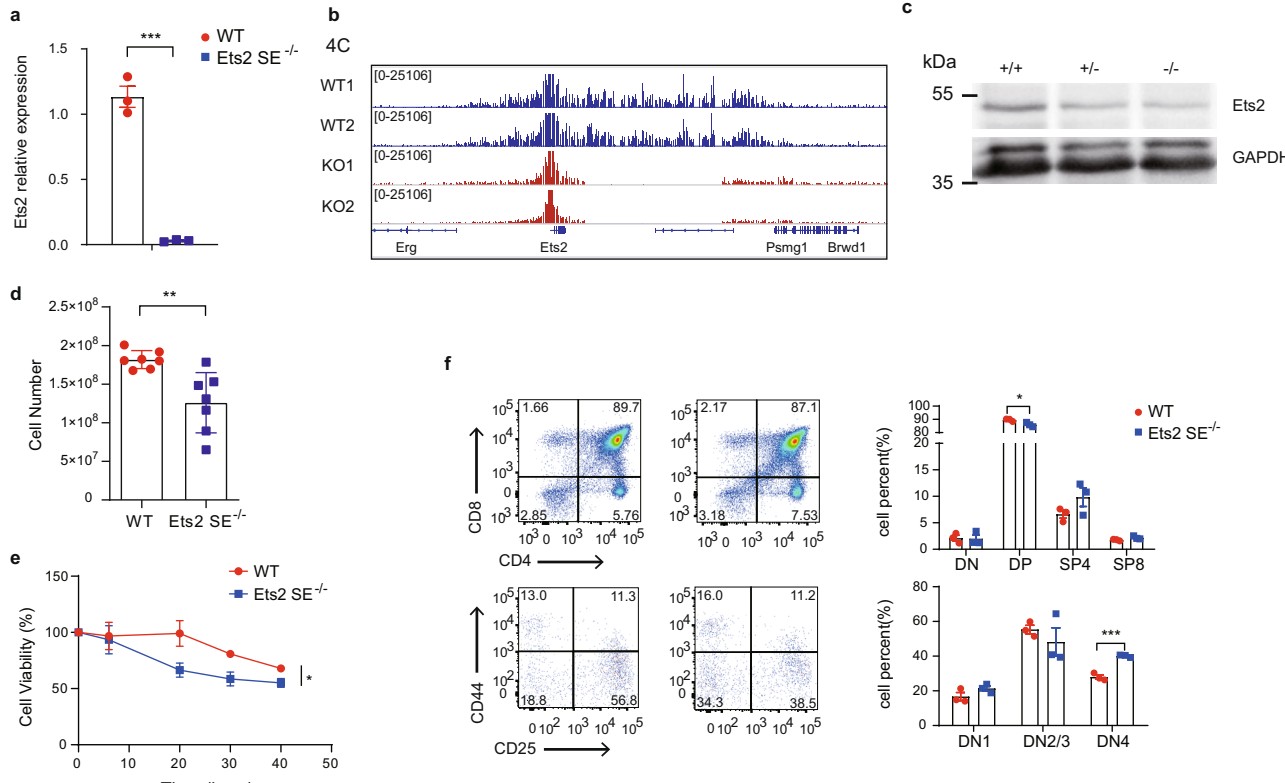

**Fig. 6 | Super-enhancer regulates *Ets2* expression in the thymus. a** Relative *Ets2* expression in thymocytes from *Ets2* super-enhancer knockout (Ets2-SE$^{-/-}$) and wild-type (WT) mice was analyzed using reverse-transcription qPCR. The expressions are normalized to the *Actb* gene and then to WT. The data represent the mean ± SD of three experiments. ***$P < 0.001$ ($P = 0.0002$) by two-sided Student's *t* test. **b** 4C assay from the viewpoint of the *Ets2* promoter in thymocytes from WT or Ets2-SE$^{-/-}$ mice. The data are presented as reads per million mapped reads (RPM). **c** Western blot showing Ets2 protein expression in the thymus of WT(+/+), Ets2-SE KO heterozygous (+/−), and Ets2-SE KO homozygous (−/−). GAPDH was used as an internal control. **d** Comparison of wild-type and Ets2-SE knockout total thymocytes. Results are the mean ± SD of seven WT and KO mice. **$P < 0.01$ ($P = 0.0033$) by two-sided Student's *t* test. **e** Thymocytes were cultured from 0 to 40 h in RPMI 1640 medium with 10% FBS at 5% $CO_2$ and 37 °C. MTS assay was performed to detect cell viability. Each experiment was repeated three times, and the data are the mean ± SD of three WT and KO mice. *$P < 0.05$ ($P = 0.0257$) by two-way ANOVA. **f** Flow cytometry of thymocytes from WT and Ets2-SE$^{-/-}$ mice. Right, the average frequency of thymocyte subsets. Results are the mean ± SD of three WT and KO mice. *$P < 0.05$ ($P = 0.032$), ***$P < 0.001$ ($P = 0.0007$) by two-sided Student's *t* test. Source data are provided as a Source Data file.

from sorted DP cells and did 3C-HTGTS assay with a bait for the *Ets2* or *Bcl6* promoter, respectively. 3C-HTGTS data revealed that the *Bcl6* promoter had broad interactions with around 1 Mb upstream region, and interactions were dense in the SE region (Fig. 5c). The interactions with the SE region reduced dramatically in SATB1-deficient thymocytes, while the interactions with the distal CTCF binding sites remained the same level (Fig. 5c). Interactions of the *Ets2* promoter were concentrated in the downstream SE region and interactions reduced substantially in SATB1-deficient cells (Fig. 5d). The H3K27 acetylation of the super-enhancers and promoters also reduced dramatically (Fig. 5c, d), indicating reduced chromatin interactions disturbed activities of super-enhancers. RNA-seq, qPCR, and western blot experiments showed the reduced expression of *Bcl6* and *Ets2* in Satb1-deficient thymocytes (Figs. 1c, 5e, f and Supplementary Fig. S1c). Analysis of *Bcl6* and *Ets2* expression during thymocyte development showed that these two genes expression peaks are in DP stages, a similar pattern as Satb1 (Fig. 5f). These results suggested that the specifically high expressions of *Bcl6* and *Ets2* in DP thymocytes were regulated by super-enhancers and SATB1.

### Super-enhancer regulates *Ets2* expression in the thymus

To explore the role of the super-enhancer in *Ets2* (named as Ets2-SE) expression in thymocytes, we generated *Ets2*-SE knockout mice in which the 166 kb region containing the *Ets2*-SE was deleted

(Supplementary Fig. 7a). *Ets2*-SE KO mice have no visible abnormal phenotypes. The *Ets2*-SE deletion dramatically reduced the *Ets2* expression in thymocytes (Fig. 6a). The 4C assay showed that the *Ets2* promoter had much less interactions with the whole locus, including upstream and downstream regions (Fig. 6b). Western blot experiment confirmed the reduced protein expression (Fig. 6c). The result indicated that the SE regulates *Ets2* expression in thymocytes.

Then, we analyzed the thymocyte development of the *Ets2*-SE deleted mice. The cell numbers of thymi reduced by 30% ($n = 7$) in the *Ets2*-SE$^{-/-}$ mice (Fig. 6c). Flow cytometric analysis showed that *Ets2*-SE deletion slightly increased the percentage of CD4$^-$CD8$^-$ double-negative 4 (DN4) thymocytes and reduced DP thymocytes (Fig. 6d), indicating the defective transition from DN to DP. We analyzed the cell viability of thymocytes in culture, and the result showed that *Ets2*-SE deleted thymocytes had a shorter lifespan (Fig. 6e), which might explain the defective development. It was reported that short lifespan of DP thymocytes caused impaired *Tcra* rearrangement[9,10]. We analyzed *Tcra* rearrangement using a single primer pair targeting C region of the *Tcra* gene during 5'rapid amplification of cDNA ends (5' RACE)[61]. MiXCR immune repertoire analysis program was used for Jα and Vα usage[62]. The *Ets2*-SE deletion did not affect the Jα and Vα usage (Supplementary Fig. 7a, b). These results indicated that the super-enhancer controls *Ets2* expression in thymocytes, which plays a role in the DN-to-DP transition and DP lifespan during thymocyte development.

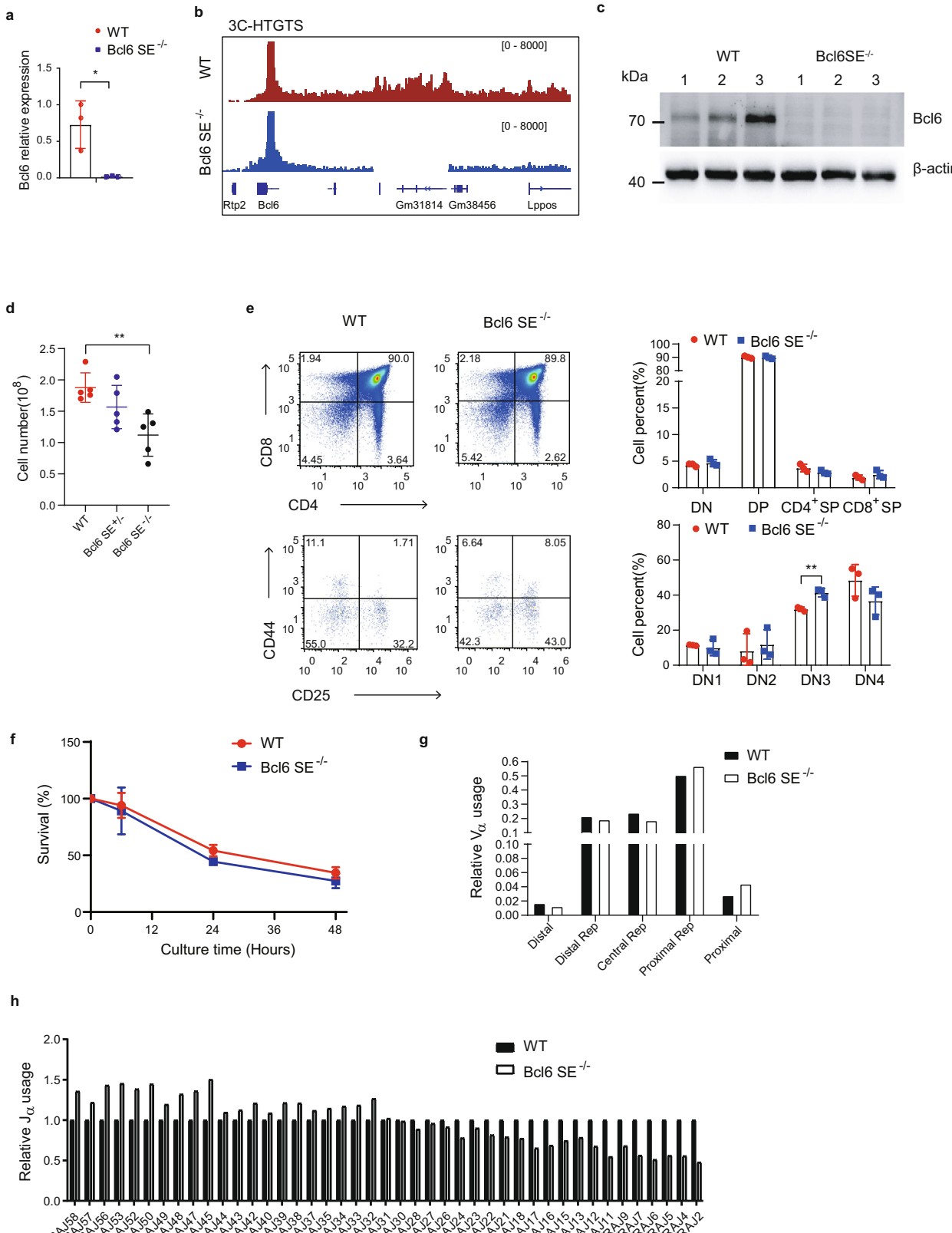

## The *Bcl6*-SE regulated *Bcl6* expression and *Tcra* rearrangement in thymocytes

To confirm the role of the *Bcl6*-SE in *Bcl6* regulation in thymocytes, we deleted a 119.2 kb region (chr16: 24,146,914–24,266,171) containing the *Bcl6*-SE in mice (Supplementary Fig. 7d). We did not observe any abnormal phenotypes in the *Bcl6*-SE KO mice. The deletion reduced the *Bcl6* expression around fifty folds in thymocytes and dramatically changed the chromatin conformation of the locus (Fig. 7a). The cell numbers of thymocytes decreased in *Bcl6*-SE homozygous mice (Fig. 7b). The proportion of DN, DP, and SP populations was not affected by the *Bcl6*-SE deletion (Fig. 7c). Within the DN population, the percentage of DN3 was increased significantly (Fig. 7c), which is

**Fig. 7 | Super-enhancer regulates *Bcl6* expression in the thymus. a** Relative *Bcl6* expression in thymocytes from *Bcl6* super-enhancer knockout (Bcl6-SE$^{-/-}$) and wild-type (WT) mice was analyzed using reverse-transcription qPCR. The expressions are normalized to the *Actb* gene and then to WT. The data represent the mean ± SD of three experiments. *$P < 0.05$ ($P = 0.0193$) by two-sided Student's *t* test. **b** 3C-HTGTS analysis of thymocytes from *Bcl6* super-enhancer knockout (Bcl6-SE$^{-/-}$) and wild-type (WT) mice. The bait is in the *Bcl6* promoter. The data are presented as reads per million mapped reads (RPM). **c** Western blot showing Bcl6 protein expression in the thymus of WT and Bcl6-SE KO homozygous (−/−). β-actin was used as an internal control. Three biological replicates are shown here. **d** Comparison of WT, Bcl6-SE KO heterozygous, and homozygous total thymocytes. Results are the mean ± SD of five mice (6–8 weeks) for each genotype. **$P < 0.01$ ($P = 0.0033$) by two-sided

Student's *t* test. **e** Flow cytometry of thymocytes from WT and Bcl6-SE$^{-/-}$ mice. Right, the average frequency of thymocyte subsets. Results are the mean ± SD of three WT and KO mice. **$P < 0.01$ ($P = 0.0035$) by two-sided Student's *t* test. **f** Thymocytes were cultured from 0 to 48 h in RPMI 1640 medium with 10% FBS at 5% $CO_2$ and 37 °C. MTS assay was performed to detect cell viability. Each experiment was repeated three times, and the data are the mean ± SD of three WT and KO mice. **g** Relative Vα and **h** Jα usages were determined by deep-sequencing of *Tcra* transcripts amplified by 5'RACE of WT and Bcl6-SE$^{-/-}$ thymocytes, respectively. The relative Vα or Jα usages were calculated by dividing the number of the clonotypes containing the Vα or Jα genes by the total clonotype number. The Vα usage is the sum of the usage frequency of all Vα genes in the region.

consistent with the observation in the conditional *Bcl6* knockout mice with a *lck*-cre transgene[59]. We also analyzed T cells in the spleen, mesenteric lymph nodes, inguinal lymph nodes, and auxiliary lymph nodes (Supplementary Fig. 8a, b). We only observed the increased ratio of CD4$^+$/CD8$^+$ T lymphocytes in inguinal lymph nodes of *Bcl6*-SE-deleted mice (Supplementary Fig. 8a, b). The *Bcl6*-SE deletion did not affect cell lifespan (Fig. 7d).

We also detected *Tcra* rearrangement using 5′ RACE sequencing. We noticed that the usages of the proximal Vα genes increased slightly and the usages of the distal Vα genes reduced in thymocytes of *Bcl6*-SE mice (Supplementary Fig. 8c). To show the difference more clearly, we combined Vα genes into five groups: the proximal, the proximal repeats, the central repeats, the distal repeats, and the distal. The usages of the proximal and the proximal repeat increased and the central repeat, the distal repeat, and the distal decreased (Fig. 7e). Consistent with the abnormal usage of the Vα genes, the Jα usage data showed an abnormal pattern with an increased 5′ Jα and reduced 3′ Jα usage in the *Bcl6*-SE deleted thymocytes (Fig. 7f). Taken together, the results indicated that the super-enhancer regulates *Bcl6* expression and plays a role in normal T-cell development and *Tcra* rearrangement.

## Discussion

DP cells are at a critical stage of T-cell development, which mainly undergo two biological processes: (1) generate highly diverse T-cell receptors through *Tcra* rearrangement and perform positive and negative selection simultaneously; (2) determine the direction of differentiation according to the TCR signal, producing CD4$^+$ SP, CD8$^+$ SP, T$_{reg}$, and iNKT, respectively. Some transcription factors such as TCF-1[63,64], E proteins[65], c-Myb[66,67], and RORγt[9], are involved in regulating DP cells. However, most of the factors only participate in one of biological processes. SATB1 regulates thymocyte development[23], *Tcra* rearrangement[6], positive and negative selection[27], and lineage decision in DP cells[25], which makes it as a versatile regulator of DP thymocytes. Here we provided evidence that SATB1 controls the DP cell identity in a single-cell transcription profile, although SATB1-deficient DP cells still highly express CD4 and CD8. Furthermore, the regulatory effect of SATB1 on DP cell identity is specific because our single-cell RNA-seq data showed that *Satb1* deletion does not change the transcription programs of thymocytes at other stages such as DN2/3 and CD4$^+$/CD8$^+$ SP.

Since the concept of super-enhancer was proposed, many studies have supported the role of super-enhancers in regulating cell identity genes[40,68]. Our data also showed that super-enhancers control genes involved in *Tcra* rearrangement and positive/negative selection. It was reported that SATB1 acts as a pioneer molecular in establishing T$_{reg}$ cell-specific super-enhancers[28]. However, SATB1 has a high occupancy at super-enhancer regions and mediates interactions within super-enhancers and between super-enhancers and promoters in DP cells, suggesting that SATB1 regulates DP cell identity genes by reorganizing super-enhancers.

SATB1 regulates many genes related to DP thymocyte function, including *Rag1*, *Tcra*, *Cd4*, *Cd8a*, etc. Here we showed that

transcription factors Bcl6 and Ets2 are controlled by SATB1 and super-enhancers in DP thymocytes. Super-enhancer knockout mice confirmed that the high expressions of *Bcl6* and *Ets2* in DP cells play a role in DP cells. Ets1 and Ets2 are members of the ETS family of transcription factors and play a role in thymocyte development, especially for DN-to-DP transition. *Ets1*$^{-/-}$ mice displayed impaired DN3-to-DP transition due to susceptibility to cell death of ND4[69]. The research on the dominant-negative truncated Ets2 transgenic mice and a phospho-mutant Ets2 (T72A) transgenic mice showed that Ets2 plays an essential role in thymocyte development[56,57]. In this study, *Ets2*-SE mice have a similar phenotype, indicating that Ets2 high expression regulated by the super-enhancer is critical for the development and survival of DP cells. Although *Ets1* is also regulated by a super-enhancer in DP thymocytes, SATB deletion did not affect its expression, suggesting that other chromatin-organizing proteins may be involved in its regulation.

Bcl6 is a transcriptional repressor that plays an essential role in the germinal center response and is also involved in leukemogenesis[52,53]. Recent studies have shown that conditional deletion of *Bcl6* lead to defective differentiation of DN to DP and abnormal activation of Notch signaling in DP cells[59]. Consistent with the previous report, *Bcl6*-SE knockout mice also displayed reduced cell number of thymocytes and defective transition from DN to DP. We also noticed that *Tcra* rearrangement was impaired, and the mechanism remains elusive. These results support the notion that SATB1 orchestrates DP thymocyte function-related genes through reorganizing super-enhancers.

In summary, this study explored the mechanism by which SATB1 controls the cell identity of DP thymocytes and provides evidence that SATB1 promotes the intra-interactions of the super-enhancers, augments the super-enhancer activity, and then enables the high expression of cell identity genes. Thus, SATB1 maintains the cell identity of DP cells and ensures the normal development of thymocytes.

## Methods

### Mouse

*Satb1*$^{fl/fl}$ *vav*-cre$^+$ mice were generated as previously described[6], and used in this study as *Satb1*-cKO mice. The *Bcl6*-SE$^{-/-}$ and *Ets2*-SE$^{-/-}$ mice were generated using CRSIPR-Cas9 system by Beijing Vitalstar Biotechnology. The deleted regions were chr16: 24,146,828–24,266,085 and chr16: 95,745,432–95,912,361, respectively. All experiments involving mice were performed using protocols approved by Southern Medical University Animal Studies Committee. Animals were housed and bred in a specific pathogen-free animal facility. The housing condition is a light-tight chamber at a constant temperature (23 ± 1 °C), humidity (55 ± 10%), and 12-h light/12-h dark (LD) cycles.

### scRNA-seq library construction and data processing

The single-cell library was constructed using the ChromiumTM Controller and ChromiumTM Single Cell 3′ Reagent Version 2 Kit (10×Genomics, PN-120237) according to the manufacturer's instructions. The final libraries were sequenced using the Illumina Hiseq 4000 (BGI-Shenzhen, China).

For each sample, the cleaned data were generated by Cell Ranger (v3.0.2) (https://github.com/10XGenomics/cellranger) and filtered for the low-quality reads and unrelated sequences. The data were aligned to mouse mm10 reference genome. Data merging, thresholding, normalization, principal component analysis, clustering analysis, visualization, differential gene expression analysis, and cell cycle phases analysis were carried out in Seurat (v4.0.5) (https://satijalab.org/seurat) according to their recommended steps. In detail, cells were sorted based on the barcodes, and the unique molecular identifiers (UMIs) were counted per gene for each cell. In total, 8872–9283 (averagely 9077) cells were captured for individual libraries, and 1161–1621 (averagely 1391) genes were detected with UMIs per cell. Cells having total mitochondria-expressed genes beyond 10% were eliminated, along with cells expressing less than 500 or greater than 3000 total genes. After this, we performed global normalization using the SCTransform function in Seurat. These pre-processed data were then analyzed to identify variable genes and principal component analysis.

For further analysis, UMAP was used for dimensionality reduction. Cells were represented in a two-dimensional UMAP plane, and clusters were identified and annotated according to the previously published canonical immune markers. The cell cycle phase score was calculated for each cell using the Seurat function CellCycleScoring. The significance of differential expression was calculated using the Wilcoxon rank-sum test. FindMarkers function in Seurat was used for the identification of differentially expressed genes with default conditions (logfc.threshold = 0.25, test.use = "wilcox", min.pct = 0.1, min.diff.pct = -Inf, verbose = FALSE, only.pos = FALSE, max.cells.per.ident = Inf, random.seed = 1, latent.vars = NULL, min.cells.feature = 3, min.cells.group = 3, pseudocount.use = 1).

## RNA isolation and bulk RNA-Seq

The sorted DP cells were isolated using a TRIzol (Thermo Fisher Scientific, 15596018). The total RNA was quantified and qualified by Agilent 2100 Bioanalyzer and NanoDrop2000. In total, 1 µg of total RNA was used for the following library preparation. The poly(A) mRNA isolation was performed using Poly(A) mRNA Magnetic Isolation Module. First-strand cDNA was synthesized using ProtoScript II Reverse Transcriptase and the second-strand cDNA was synthesized using Second Strand Synthesis Enzyme Mix (NEB, E7550L). The purified double-stranded cDNA was subjected to end repair, 3′-dA tailing, and adapter ligation. Size selection of adapter-ligated DNA were performed prior to PCR amplification. The ligated DNA was then amplified by 10–15 cycles with Illumina P5/P7 primers. The PCR products were cleaned up using AMPure XP beads (Beckman Coulter), validated using an Qsep100 and quantified by Qubit3.0 Fluorometer (Invitrogen). Libraries were a 2 × 150 bp paired-end (PE) sequenced on a NextSeq 550 on an Illumina HiSeq instrument. Three biological replicates were performed in Satb1WT and Satb1cKO DP T cells.

Fastq files were processed by Cutadapt (v1.18) (https://github.com/marcelm/cutadapt) to be high-quality clean data, then clean data were aligned to mouse genome (mm10) by Hisat2 (v2.2.1) (https://github.com/DaehwanKimLab/hisat2). The abundance or the coverage of each transcript was determined by read counts and normalized using the number of reads per kilobase exon per million mapped reads (RPKM). Genome browser tracks in bigWig format were generated using bamCoverage function of deeptools (v3.5.0) (https://deeptools.readthedocs.io/en/develop).

Differential expression genes analysis was performed using DESeq2 (v1.30.0) (https://github.com/mikelove/DESeq2) with default setting. The increased genes were defined as log2FC > 1 and adjusted $P < 0.05$, and the decreased genes as log2FC < −1 and adjusted $P < 0.05$. To verify the rationality of the DEGs, the volcano plot and heatmaps were draw using ggplot2 (3.3.5) (https://github.com/tidyverse/ggplot2) in R. To cluster the samples and calculate the correlation

coefficients between the samples, a Spearman correlation test was applied, and the results was visualized using R package pheatmap (v1.0.12) (https://github.com/raivokolde/pheatmap).

Using WebGestalt (http://www.webgestalt.org) database on DEG set to obtain all Gene Ontology (GO) terms and KEGG pathways, accompanied by the number of genes in that GO-term and pathway, enriched $P$ value, and FDR. Only the GO-term and pathway with an FDR value <0.05 were considered as significantly enriched.

RNA-seq data from mouse T-cell precursors in different developmental stages, including DN1, DN2a, DN2b, DN3, and DP, CD4⁺SP, CD8⁺SP (Gene Expression Omnibus accession: GSE109125) were used to create DN1 and DN3 gene sets. Of the RNA-seq datasets, genes that were differentially upregulated ($P < 0.05$ and Log2 fold change (FC) > 1) between DN1 *versus* DP, DN3 versus DP were used as gene sets for GSEA. GSEA was run on all expressed WT_DP *versus* Satb1cKO_DP RNA-seq genes, which were ranked by log2FC value.

Some DEGs were confirmed by quantitative PCR with primers: *Il2Ra*: 5′-CACTACGAGTGTATTCCGGGA-3′ and 5′-TCGGTGGTGTTCTCTTTCATCT-3′; *Nme4*, 5′-GGGACTGTGATACAACGCTTT-3′ and AGAGCTGGGTAGAATGGCTTC-3′; *Slc35f2*, 5′-CCATCACCAGCCAGTATTTGG-3′ and 5′-GCATCAACGTGTAAACCAGGA-3′; *Mnd1*, 5′-AGAGAACCCGGATGATGGAGA-3′ and 5′-CTGACATGGCGGTTATGCCTT-3′; *Rnf144a*, 5′-TGCTGGTACTGCCTGGAGT-3′ and 5′-ATGCCAGATCACTGATGCTCG-3′; *Cdk6*: 5′-TGGACATCATTGGACTCCCAG-3′ and 5′-TCGATGGGTTGAGCAGATTTG-3′; *Gapdh*: 5′-AATGGATTTGGACGCATTGGT-3′ and 5′-TTTGCACTGGTACGTGTTGAT-3′. The total RNA was extracted using Trizol (Thermo Fisher Scientific, 15596018), cDNA synthesis by HiScript III All-in-one RT SuperMix Perfect for qPCR mix (Vazyme, R333-01). RNA expression was normalized to Gapdh expression. The antibodies used for western blot are anti-SATB1 (1:2000 dilution, rabbit monoclonal [EPR3951], Abcam, ab109122), anti-Bcl6 (1:400 dilution, Mouse monoclonal [GI191E], Abcam, ab241549), anti-ETS2 (1:400 dilution, Rabbit monoclonal [EPR22419], Abcam, ab219948), anti-actin (1:4000 dilution, FD0060, Fdbio science, China), and anti-GAPDH (1:4000 dilution, FD0063, Fdbio science, China) antibodies.

## ChIP-seq

ChIP-seq data were either generated in this study or downloaded from a public resource. The raw data were processed and analyzed according to the following procedure. First, SRA files were converted to fastq format, then aligned to mouse genome (mm10) using Bowtie2 (v2.3.5.1) (Bowtie2 (v2.3.5.1) (https://github.com/BenLangmead/bowtie2). PCR duplicated fragments were filtered by Picard (v1.118) (http://broadinstitute.github.io/picard). Then, filtered reads were mapped. Peaks were identified by Homer (v4.10.4) (https://github.com/bastienwirtz/homer). FRiP (Fragments Ratio in Peaks) value was calculated using bedtools (v2.29.2) (https://github.com/arq5x/bedtools). We used deepTools (v3.5.0) (https://github.com/deeptools/deepTools) to generate bigWig file with RPKM normalization. The enriched peaks region was used as input to DESeq2 (v1.30.0) (https://github.com/mikelove/DESeq2) to find differential peaks from ChIP-Seq data as well as normalized the data. Peaks were annotated by using R package ChIPSeeker (v1.26.0) (https://github.com/YuLab-SMU/ChIPseeker). For mapping peaks to gene features, we identified distribution of the peaks of each ChIP-Seq data across the genome. The promoter was defined as a region within ±3 kb from the TSS, and peaks without being mapped to promoter, upstream, intron, or exon were considered as intergenic target loci. Profiles were obtained on a region of ±3 kb from the center of peaks, and average scores were plotted to generate averaged read density around peaks using ngsplot (v2.41) (https://code.google.com/p/ngsplot/).

## Identification of super-enhancers

Super-enhancers (SEs) were identified using the rank ordering of super-enhancers (ROSE) algorithm (https://github.com/stjude/ROSE). H3K27ac peaks were used to define enhancers, followed by further

filtering based on the criteria: briefly, peaks located within ±3 kb region of TSSs were excluded. The remaining H3K27ac peaks were defined as putative enhancers. Enhancers located within ±12.5 kb regions of each other were stitched together, scored, and ranked based on H3K27ac ChIP-Seq signals. Enhancers were plotted with enhancer rank versus enhancer density, and all enhancer regions above the inflection point of the curve were defined as SEs. An analogous procedure was used to define SE regions by enrichment of Satb1 at enhancers. Super-enhancers and typical enhancers were assigned to the genes using the default parameters of the ROSE algorithm. ChIP-seq signal at SE regions were plotted as averaged profiles in ngsplot. GSEA was used to identify how SE-gene sets distribute in gene lists ranked by either gene expression fold change values or H3K27ac ChIP-seq enrichment on enhancers. GO analysis for SE-associated genes was performed with WebGestalt (http://www.webgestalt.org) database.

## Hi-C

We performed in situ Hi-C with 5–10 million cells. Cells were cross-linked at a final concentration of 1% formaldehyde and in ice bath for 10 min and quenched by 200 mM glycine for 5 min at room temperature. After two washes with cold 1×PBS, cells were pelleted and kept at −80 °C until use. Subsequently, cross-linked cells were lysed, and chromatin was digested with 150 U MboI (NEB, R0147L) overnight at 37 °C. MboI was inactivated at 62 °C for 20 min, DNA fragment ends were biotinylated with biotin-dCTP using Klenow large fragment (NEB, M0210V) for 60–90 min at 37 °C, and the sample was diluted, and proximity ligated for 4 h at room temperature. Cross-linked DNA was reversed by the addition of SDS, proteinase K, and NaCl, and allowed to incubate overnight at 68 °C. DNA was purified by phenol/chloroform, followed by ethanol precipitation, and resuspended in 100 μl nuclease-free dH₂O. DNA was treated with T4 DNA polymerase (NEB, M0203L) to remove unligated biotinylated ends and sheared to 300–500 bp by sonication subsequently, the ligated junctions were pulled down with Dynabeads MyOne Streptavidin C1 magnetic beads (Thermo Fisher Scientific, 65001). End repair, dA tailing, adapter ligation, and PCR amplification were performed on biotinylated DNA fragments bound to beads. After purification, libraries were sequenced on an Illumina HiSeq 2000 platform to obtain 150 bp paired-end reads.

For each sample, reads were obtained following quality filtering and adaptor trimming using fastp (v0.20.0) (https://github.com/OpenGene/fastp). Hi-C mapping, filtering, correction, and binning were performed with Hi-C-Pro (v2.11.1) (https://github.com/nservant/Hi-C-Pro). The paired-end reads were mapped to the mouse mm10 genome. Self-circle ligation, dangling ends, re-ligation, and the other dumped types were filtered out with Hi-C-Pro after mapping. We generated raw contact matrices at 10 kb, 20 kb, 50 kb, 100 kb, 500 kb, 1 Mb resolutions. For raw contact matrix correction, we used the iterative correction method (ICE). The hicpro2juicebox.sh utility was used to convert the allValidPairs output of the pipeline into Juicebox hic format at fragment resolution. Visualization of Hi-C contact matrices was done with Juicebox (v1.5.1) (https://github.com/aidenlab/Juicebox). Differential analysis and visualization of local interactions from Hi-C data were obtained using HiTC (v1.34.0) (https://github.com/bioinfo-pf-curie/HiTC) R package.

A/B compartment analysis was performed at 250 kb resolution using a publicly available script matrix2compartment.pl (https://github.com/dekkerlab/cworld-dekker). Then, we used the first principal component (PC1) to predict regions of active (A compartments) and inactive chromatin (B compartments). We generated 250-kb tracks and correlated A/B compartments with H3K27ac and H3K27me3 ChIP-Seq data for each cell type using the H3K27ac mark as an indicator of A compartments and the H3K27me3 mark as an indicator of B compartments. We identified the regions with changes (A-A, B-B, and A-B/B-A) in sign of the PC1 value between Satb1WT and Satb1cKO DP cells as A/B compartment switched regions.

The TAD structure (insulation/boundaries) was defined by the insulation score. The matrices which were used to calculate the insulation score were normalized by ICE method for discarding the bias of raw matrices. And then, the insulation score was computed at 10 kb resolution. The script matrix2insulation.pl was used to detect TAD boundaries, with the following options: '--is 600000 --ids 250000 --im mean --bmoe 3'. TADs were called using normalized Hi-C matrices at 10-kb resolution with insulation2tads.pl. The script can be accessed through GitHub (https://github.com/dekkerlab/cworld-dekker).

We generated a chromatin interaction matrix with a 50 kb resolution using the Hi-C data, and the differential chromatin interactions between Satb1WT and Satb1CKO DP cells were calculated using the Bioconductor package multilHiCcompare (v1.8.0) (https://github.com/dozmorovlab/multiHiCcompare), which provides functions for the joint normalization and detection of differential chromatin interactions in our two Satb1WT and two Satb1CKO replicates experiments. To identify the significantly differential chromatin interactions, we used an adjusted $P$ value of 0.05 and a log fold change of 0. The chromosomal sequencing was divided into 50-kb bins for the normalized signal file of the Satb1WT ChIP-Seq. Once the bin attained by ChIP-Seq was located within the differential chromatin interactions, the value in the corresponding bin of the Hi-C matrix indicated the normalized count.

Chromatin loops were called using Fit-Hi-C (v2.0.7) (https://github.com/ay-lab/fithic). Input files of Fit-Hi-C were created by using a publicly available script hicpro2fithic.py from Hi-C-Pro. Next, loops were called using fixed-size bin resolutions from 10 to 20 kb in both cell types. Briefly, significant interaction loops ($P <= 0.05$ and contact frequencies $>= 5$) were identified through jointly modeling the contact probability using raw contact frequencies and ICE normalization vectors with the Fit-Hi-C algorithm. Enhancer–promoter loops were annotated using pgltools (v2.2.0) (https://github.com/billgreenwald/pgltools). The anchors of loops were intersected with promoters and enhancers. Promoters were defined as ±5-kb windows of the TSS of all expressed genes, enhancers are defined using the enhancer dataset of DP cells (http://www.enhanceratlas.org/downloadv2.php). Super-enhancer loops were defined by H3K27ac-definded SE sets obtained through ROSE software.

The format of Hi-C data,.hic, was converted into.cool files using hic2cool (v0.8.3) (https://github.com/4dn-dcic/hic2cool). Hi-C matrices in cool format were used to generate genome-wide aggregate plots at TADs and loops of Satb1WT and Satb1cKO DP cells detected by Hi-C. We used coolpuppy (v0.9.5) (https://github.com/open2c/coolpuppy) to pile-up normalized Hi-C signals at a 10-kb resolution at loops previously identified and plotted 500 kb upstream and downstream of the loop anchor coordinates. The local rescaled pileups of TADs annotated using insulation score valleys used above in DP cells were analyzed at a 10-kb resolution. We plotted them using plotpup.py of coolpuppy.

## 4C

Chromatin was cross-linked for 10 min at room temperature in 1 × PBS/10% FBS containing 4% formaldehyde. Cross-linking was blocked by glycine addition. Following cell lysis, nuclei pellets were resuspended in 1.2×Buffer2 (NEB, B7002S), followed by SDS addition. The samples were incubated while shaking for 60 min at 37 °C. Triton X-100 was added to quench SDS. Cross-linked chromatin was primarily digested using MboI (NEB, R0147L). Ligation was performed in the presence of T4 DNA Ligase (NEB, M0202L) in diluted conditions. Chromatin reverse cross-linking was carried out using proteinase K, phenol-chloroform and DNA were purified by ethanol precipitation. Secondary restriction was performed using NlaIII (NEB, R0125L). Secondary ligation was carried out with T4 DNA Ligase in diluted conditions. Ligated DNA products were then extracted using phenol-chloroform with ethanol precipitation and purified. 4C-seq library preparation was achieved by inverse PCR using 100 ng template DNA (for ten reactions in total). The viewpoint-directed inverse PCR primers carrying Illumina

P5 or P7 sequencing adapters as below: Ets2 forward primer: 5′-GTGACTGGAGTTCAGACGTGTGCTCTTCCGATCTgcccgggagcagcgtggatc-3′; Ets2 veserce primer: 5′-TCTTTCCCTACACGACGCTCTTCCGATCTcgagcacttaggcgccatct-3′.

## 3C-HTGTS

3C-HTGTS libraries were prepared with FACS-sorted DP cells. In brief, 10 million cells were cross-linked with 1% formaldehyde at room temperature for 10 min, quenched with glycine (final concentration 0.125 M) on ice for 5 min. Cell were lysed and followed by the addition 200U MboI to digest the chromatin overnight at 37 °C with gentle shaking. MboI was inactivated by adding 10% SDS to a final concentration of 1.5% and incubating at 37 °C for 30 min. To reduce the SDS concentration, the solution was diluted with T4 ligase (NEB, M0202L) buffer containing 1% Triton X-100, followed by incubation at 37 °C for 1 h. T4 ligase (New England Biolabs) was added and incubated for overnight at 16 °C. Cross-linking were reversed and samples were treated with proteinase K and RNase A prior to DNA extraction with 1:1 phenol-chloroform and precipitation with ethanol. The 3C libraries were sonicated to 300–500 bp on a Qsonica Bioruptor Sonicator. Sonicated DNA was linearly amplified with a biotinylated primer (Bcl6: 5′-TTACCATTGCTCCGCAGCAG-3′, Ets2: 5′-GGACCTGCAGACAGCCTAAC) that anneals the promoters of Bcl6 or Ets2. The biotin-labeled single-stranded DNA products were enriched with streptavidin C1 beads (Thermo Fisher Scientific, 65001), and followed by 3′ ends ligation with the bridge adaptor. The adaptor-ligated products were amplified through nested PCR using a nested primer and an adaptor-complementary primer (Nest primers: Bcl6: 5′-CTGGAGTTCAGACGTGTGCTCTTCCGATCTTCCCGGCCGGCAGAATGCCT-3′, Ets2: 5′-CTGGAGTTCAGACGTGTGCTCTTCCGATCTAGCTGCAGGGTCGCAGAGAA-3′ and an adaptor-complementary primer: 5′-TCTTTCCCTACACGACGCTCTTCCG ATCTGACTATAGGGCACGCGTGG-3′). And a final PCR for another 10–15 cycles of amplification with P5 and P7 was performed. After purification, the final libraries were sequenced on an Illumina HiSeq 2000 platform to obtain 150 bp paired-end reads.

3C-HTGTS fastq data were filtered by removing adaptor, and low-quality reads with using fastp. Filtered reads were extracted from the sequence file after quality control through Cutadapt and Pear (v0.9.6) (https://github.com/tseemann/PEAR). Pair-end reads containing Nest-Primer or AdapterPrimer were obtained, and then the reads were filtered by searching restriction sites sequences. The remaining reads were mapping to mouse genome mm10 with bowtie2 (v2.3.5.1). The mapping reads were filtered the duplicated reads, self-ligation reads, relegation reads, and dumped reads. For visualization, we convert the final bam files into bedGraph file. The signal peak bedGraph file was obtained by post-comparison filtering, signal statistics, and standardization. We normalized bedGraph file using CPM (Counts Per Million in cis) normalization method, and visualized on IGV. Finally, we organized the results report and visualized it with R.

## FACS analysis

Thymus, spleen, mesenteric lymph nodes, inguinal lymph nodes, and auxiliary lymph nodes from 6 to 8-week-old mice were ground in MACS buffer (1× PBS, 0.5% BSA, 2 mM EDTA) and filtered with 40-μm nylon mesh. Red blood cells were lysed in RBC buffer (Biolegend, 420301) for 10 min at room temperature. FACS analysis was performed on the BD FACS Aria3 machine. DP cells were defined as CD4$^+$CD8$^+$. Cells were gated on CD4$^-$CD8$^-$ and DN cells were defined as followed: DN1 CD44$^+$CD25$^-$, DN2 CD44$^+$CD25$^+$, DN3 CD44$^-$CD25$^+$, and DN4 CD44$^-$CD25$^-$. The T lymphocytes were defined by the CD3 expression. Cells from the spleen, mesenteric lymph nodes, inguinal lymph nodes, and auxiliary lymph nodes were gated with CD3, and the mature T cells were defined as CD4$^+$ or CD8$^+$. Antibodies used for FACS: anti-CD3-EFLUOR 450 (eBioscience 48003280, Clone 17A2), anti-CD25-PE (Biolegend 102008, Clone PC61), anti-CD8a-FITC (Biolegend 100706,

Clone 53-6.7), anti-CD4-APC (Biolegend 100516, Clone RM4-5), anti-CD44- PE/Cy5 (Biolegend 103010, Clone IM7). Cells were stained at a concentration of 0.25 μg antibody per million cells in 100 μl FACS buffer (1× PBS, 2 mM EDTA, and 1% BSA). FACS data were analyzed using CellQuest Pro software (BD Biosciences).

## 5′ RACE sequencing

The 1 μg RNA was used for the reverse transcription (RT). The cDNA libraries were performed with oligo(dT) primer by 5′ Rapid amplification of complementary DNA ends(5′RACE). During the reverse-transcription reaction, A SMART oligonucleotide (5′-AAGCAGTGG-TATCAACGCAGAGTACGCrGrGrG) was ligated to the 5′end of each cDNA. The TCR library is then prepared by touchdown PCR amplification using a set of primers: a forward primer to the adaptor and a reverse primer to the constant region of the TCR. Touchdown PCR was performed under the conditions of 95 °C for 3 min, followed by 15 cycles of 98 °C for 15 s, 65 °C (decrease 0.7 °C/cycles) for 15 s and 72 °C for 1 min, 20 cycles of 98 °C for 15 s, 60 °C for 15 s and 72 °C for 1 min and one cycle 10 min at 72 °C. PCR products, ranging from 300 to 700 bp were gel purified by DNA Gel Purification Kit. Illumina paired-end adapters were ligated to TCR libraries, followed by second-round PCR with Illumina P5/P7 adaptor primers under 30 s at 95 °C, 15 s at 65 °C, 60 s at 72 °C for 10–15 cycles, plus a final extension for 10 min at 72 °C. Ampure bead purified PCR product. Finally, the TCR libraries were sequenced on an Illumina HiseqXtenPE150 (2 × 150 nt).

Single-cell pellets of thymus were subjected to RNA extraction using phenol and chloroform. In total, 500 ng mRNA were then reverse transcribed according to previous reports. Next, Tcra and Tcrb sequencing amplicons were amplified using the primers described below: forward primer: 5′-TGAACCTTAAGCAGTGGTATCAACGCAGAG, Tcra: 5′-CAGGGTCAGGGTTCTGGATA-3′, Tcrb: 5′-TCTGATGGCTCAAACACAGC-3′. The library for Illumina sequencing was prepared using DNA Library Prep Kit for Illumina (NovoNGS, Shanghai, China).

The 5′RACE raw data were filtered by fastp and the adapter sequences, contamination, and low-quality reads were removed. T-cell receptor beta chain V, D, and J gene identification, CDR3 sequence extraction in clean reads were performed using MiXCR (v3.0.11) (https://github.com/milaboratory/mixcr). The corresponding germline sequences were mapped to the reference sequences derived from international ImMunoGeneTics (IMGT) information.

### Reporting summary

Further information on research design is available in the Nature Research Reporting Summary linked to this article.

## Data availability

The data that support this study are available from the corresponding authors upon reasonable request. The raw sequence data of single-cell RNA-Seq, bulk RNA-seq, ChIP-seq, Hi-C, 4C-seq, 3C-HTGTS, and 5′RACE reported in this paper have been deposited in the Gene Expression Omnibus (GEO) database under the accession number: GSE182995. GEO accession codes (or SRA accession number) of the published data used in this study are as follows: H3K4me3 ChIP-seq of CD4 + CD8 + DP thymocytes GSE21207; Rad21 and Nipbl ChIP-seq, GSE48763; CTCF ChIP-seq, GSE141223; SATB1 ChIP-seq, GSE90635; RNA-seq of T-cell development, GSE109125; ATAC-seq of T-cell development,GSE100738; H3K-27ac and H3K4me1 ChIP-seq of Satb1WT and Satb1cKO DP thymocytes, DRP003376; 3e Hi-C data, GSE79422. The remaining data are available within the article, supplementary information or available from the authors upon request. Source data are provided with this paper.

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

## Acknowledgements

We thank Michael Krangel from the Department of Immunology at Duke University for Satb1cKO mice, Hongfen Shen and Qianbing Zhang for Flow cytometry, Zhongxi Huang and Jiahong Wang for research computing. This work was supported by the National Natural Science Foundation of China (31970836 and 32170885 to B. Hao, 81801549 to L. Qin), Guangdong Province Natural Science Foundation (2022A1515010409 to B. Hao), Young talents of Health Science and Technology Innovation in Henan Province (YXKC2021002 to B. Hao), and the National Institutes of Health grant (NIH R01 ES023854 to T. Kohwi-Shigematsu).

## Author contributions

D.F. designed, performed, and analyzed the following experiments: scRNA-seq, bulk RNA-seq, Hi-C, 4C-seq, 3C-HTGTs. Y.C. performed FACS and 5′RACE of *Bcl6*-SE mice. R.D. performed bioinformatics analysis of ChIP-seq, Hi-C, 3C-HTGTs, and 5′RACE. S.B. designed the study, performed FACS and 5′RACE of *Ets2*-SE mice. W.X. performed FACS, ChIP-seq, 3C-HTGTS library preparation. Y. Zhu performed 3C-HTGTS library preparation. Z.L. performed 4C-seq, ATAC-seq and ChIP-seq analysis. J.B. performed 5′RACE library preparation. Y. Zhang provided technical support of Hi-C library preparation. Y.Y. provided technical support, reagents, and conceptual advice. J.Z. assisted with bioinformatical analysis. L.Q. provided technical support, reagents, and conceptual advice. Y.K. did control ChIP-seq experiment with SATB1 in KO thymocytes. W.S. provided technical support and conceptual advice. T.K.-S. provided *Satb1* conditional knockout mouse and did critical discussion in manuscript preparation. J.M. did an analysis of scRNA-seq data. S.L. supervised the project. B.H. wrote the manuscript with help from D.L., Y.C., R.D., S.B., X.W., and T.K. All authors read and approved the final version of the manuscript.

## Competing interests

The authors declare no competing interests.

## Additional information

[1]Cancer Research Institute, Experimental Education/Administration Center, School of Basic Medical Sciences, Southern Medical University, 510515 Guangzhou, Guangdong, China. [2]Medical Genetic Institute of Henan Province, Henan Key Laboratory of Genetic Diseases and Functional Genomics, Henan Provincial People's Hospital, People's Hospital of Zhengzhou University, Zhengzhou university, 450003 Zhengzhou, Henan, China. [3]Department of Orofacial Sciences, University of California, San Francisco, San Francisco, CA 94143, USA. [4]ENT institute, Department of Facial Plastic and Reconstructive Surgery, Eye & ENT Hospital, Fudan University, 200031 Shanghai, China. [5]National Health Commission Key Laboratory of Birth Defects Prevention, School of Medicine, People's Hospital of Henan University, Henan University, 450003 Zhengzhou, Henan, China. [6]These authors contributed equally: Delong Feng, Yanhong Chen, Ranran Dai, Shasha Bian, Wei Xue. ✉e-mail: ychslshx@henu.edu.cn; haobt123@163.com

