## [Peer Review File · Nature Communications]

REVIEWER COMMENTS

Reviewer #1 (Remarks to the Author):

Feng et al. investigated the mechanism of regulation of gene expression by SATB1 in DP thymocytes. Using scRNAseq analysis of SATB1-deficient thymocytes they show that the cell identity of DP thymocytes was changed, and the DP specific genes were down-regulated. They further argue that Satb1 regulates super-enhancer landscape of specific to DP cells while giving examples of 2 SATB1 target genes Bcl6 and Ets2 via deletion of their super-enhancers. Further, using HiC analysis the authors showed that interactions between super-enhancers and promoters decreased in SATB1 deficient thymocytes. This data extends on the previously known roles of SATB1 towards regulation of super-enhancers and establishment of DP cell identity.

The authors have used the power of sc-RNAseq technology to demonstrate the cell type/number changes in the thymus. Further, they show specific nature of DP signature genes' dysregulation upon Satb1 deficiency. Authors hypothesized and showed subtle changes in the global genome interactions upon Satb1 KO via HiC. The authors selected 2 of the Satb1 targets via SE analysis, namely Bcl6 and Ets2, generated KO models by deleting their super-enhancers, and demonstrated their individual roles. Although, the manuscript has good potential and approach by analyzing number of datasets, there are multiple aspects in which the authors need to strengthen their work. Many of their models and experiments require further validation. Additional experimental evidences are required for supporting multiple conclusions, e.g. to demonstrate if Satb1 actually loops the SE for Ets2 and Bcl6 to their respective promoters. The manuscript will also benefit from clear writing and elaboration of the results.

Specific Comments:

1. In Fig. 1a. the authors show that there is reduction of DN and SPs along with increase in DP cell numbers (with the example of cluster 3, line 102). Although there is a dramatic reduction in cluster 11 and a significant reduction in cluster 0, which are also DP clusters. Are there any differences in gene expression profiles for these clusters that drive their reduction compared to cluster 3? The authors should explain these contrasting populations both in the figure and the text.
2. Similarly, the DN4/ISP cluster 15 has disappeared in the KO UMAP, which is contradicting their certainty as an increasing population as mentioned in the text. The authors should therefore include the corresponding flow cytometry plots for each stage to verify whether it corroborates the scRNAseq data.
3. Additionally, the B cell number is dramatically increased in the KO condition as shown in cluster 7. Is there a B cell specific gene downregulation associated with Satb1 KO?

4. How correlative is the scRNAseq data from DP bulk RNA-seq data? The authors should discuss this as the gene expression profiles are not always similar in scRNAseq data. Further validation would be required.

5. The text (line 106) does not correlate with the associated figure S2, as there is opposite trend shown for up-and-down genes. The authors should carefully check for such inconsistencies. Further, for S2e it seems unclear what the authors wish to conclude from the same. Which gene

6. In Fig. 1d, the authors posit that downregulated genes are mostly from the DP stage, whereas the upregulated genes belong to the prior stages. It seems that they have generated the heatmap from the scRNA expression data from all stages, but it is unclear both in the text and legend. If it is the bulk RNA data, then this should be explained. Similarly, the results from many other figures are not clearly elaborated. The authors should explain the figures clearly throughout the manuscript.

7. The authors further claim that DN specific genes are upregulated in DPs, but the figure indicates that a similar ratio of downregulated genes is also present for the DN stages as well. How do the authors reconcile these findings?

8. Fig. 1f: The authors state that Satb1 mediated repression is tested using GSEA with DN1/2 gene sets. It is not clear in the figure/text which genes they have used as their ranked list for the statistic. It is all the more required since using up- or down- regulated genes would change the interpretation of the leading edge of the plot, hence the conclusion. These findings should be validated by quantifying the gene expression using PCR assays.

9. Fig. 2, the super-enhancer landscape is thought to be established a priori the stage in which the gene expression is controlled. The authors should also monitor the super-enhancers in DN4/ISP stages and how the expression of those genes are affected in DP. Importantly, as many of the super-enhancer gene expression show increased correlation with DP super-enhancers (shown in 2b) in previous stages, the authors should experimentally validate these bindings and gene expression in stage-wise manner. The second sub-panel of 2b should be shifted to supplementary data.

10. In Fig. 2c, are the ChIP-seq datasets used derived from sorted DPs? Its not clear in either the figure or the text. The authors should revise their text to be more elaborative since a single stage is in question and the conclusions are drawn for specificity of Satb1 function in DPs. Furthermore, the authors should include experimental data for the mined data to ascertain its correlation with their own datasets.

11. In the context of Fig. 2d, the authors show a modest enrichment of Satb1 at the K27ac enriched regions, however its binding seems to be opposite to the modification when they plotted the average plot around Satb1 peaks. The authors should discuss this in details as the conclusions drawn are affected. Authors also mention that there is promoter binding, but data is presented only as distance from super-enhancers and overall promoter binding (S3e), hence warrants inclusion of supporting data in form of distance tag plots.

12. In Fig. 2e, the authors show the modest decrease in K27ac occupancy plotted for SEs. Although there is a much dramatic decrease in TEs (shown in supplementary). The authors should discuss this important distinction in the text. Additionally, the authors should show the percentage of gained and lost SEs upon Satb1 KO.

13. In Fig. 2g, the authors posit that more genes are downregulated associated with SEs upon Satb1 KO. According to the figure, the authors seem to have used all SEs and not just the ~246 SEs they identified specific to DP in Fig. 1a. The authors should revise all the plots wherein they show a DP specificity to bolster their findings. 13. Are the gained SE and gene expression (2h) not 'DP specific' genes?
14. Fig. 2o, and related text in line 316 does not correlate with the supplementary data which showed that TEs are more affected than SEs in Satb1 KO. The authors should reconcile these findings.
15. Fig. 3b. What is the dataset used to rank the genes? The authors also need to show the association of SEs identified with Satb1 enriched cluster plot.
16. In Fig 3d, as the plotting is done around DP for highly dysregulated genes, the authors should also plot the heatmaps around DN4/ISP, so as to strengthen the conclusion of DP specificity of expression. Further, this plot should be moved to Fig. 1, along with the overall scRNAseq data.
17. Figure 3e. Since the authors posit that there is strong association of Satb1 with SEs of DP, the pathways very strongly enriched for Satb1 in DPs are only enriched with much lower confidence and genes in DP SEs. Can the authors also show processes more highly enriched for high confidence SE genes.
18. In Fig. 4 b, the authors plotted the densities of Satb, Ctf and K27ac for regions of lost and gained loops in the KO condition. Although it is clear that Satb1 and Ctf occupancy is more on the lost interactions, K27ac is only modestly more, which suggests its ubiquitous binding irrespective of Satb1. The authors should present detailed plots to reveal the minor differences, and the actual concordance between the lost and gained analysis among the three.
19. Fig. 4c, the authors should also plot the counts of loops in the KO condition. Its mentioned in the text but not plotted per se. Additionally the supplementary figures the authors referred to exhibit much more dramatic differences (assuming prep 1 and prep2). Since the difference in the two preps shown is quite high, the authors should discuss this in the text.
20. The authors mention in line 173, 'The loop strength increased in the promoters of the upregulated genes and decreased in the downregulated genes'. Are the increased loop strength genes of DN lineage? These analyses will further strengthen the major point of the manuscript.
21. Fig. 5, experimental evidence showing if Satb1 actually loops the SE for Ets2 and Bcl6 to their respective promoters is required for the conclusions drawn.
22. Fig. 5b and c, the 3C-HTGTS data shows much broader interaction extending on either sides from the SE (assuming its labeled red), which also decreases. The authors should discuss this in the text. This will also support the next Fig. 6b.
23. The authors further mention that ETS SE -/- results in reduced cell number by 69% (line 216) whereas Fig 6c shows at modest ~30%. These need to be confirmed.
24. According to Fig. 6d, the DN/DP ratios do not correspond with the Satb1 KO ratios. The authors should discuss this, as they claim Satb1 mediated effector function by Ets2.
25. Fig. 6d: the conditions for FACS plots are not mentioned. Same issues also applicable to Fig. 7.

26. In case of ETS2 or BCL6 SE KO, is there any change in Satb1 targets? Since it seems that sections covering Fig.s 6 and 7 are separate results in themselves, the authors should perform RNA sequencing or at least qPCRs in these conditions as well. Is there any phenotypic/cellular characteristics that match with Satb1?
27. In Figs 6 and 7, the authors show that expression of ETS2 and BCL6 decrease upon their respective SE KO. These findings should be validated further by providing protein data for KO and WT mice.
28. Since the authors generated specific mice lines, it is of utmost importance to show validate their KO models via the confirmation of deletion of genomic regions, as well as phenotypic characterization such as organ size etc.
29. Since Ets1 KO is reported previously (Eyquem et al 2004) to exhibit reduced DP and SPs as DN population increases. However, the transgenic line used here does not recapitulate the same to the full extent especially at SP level (Fig 6.) Hence, the authors need to show the differences and discuss the same in the discussion.
30. According to the data presented, the Bcl6 SE KO shows no change in thymic populations as well as recombination. Since the authors claim no role of Bcl6, they should confirm by recapitulating the findings (as supplementary) of the Bcl6^{-/-} study which showed its role in TH2/Tfh differentiation.
31. It is essential to perform Satb1 looping validation for the Ets1 and Bcl6 loci in WT vs KO condition. This will highlight one of the major new findings of this manuscript, showing the role of SATB1 in the regulation of their SE.
32. Validation experiments such as WB, imaging etc. are much needed here, since both Fig.s 6 and 7 depict SE deletion, which may or may not change its expression in a cell type/Satb1-dependent manner. Hence, WB of sorted populations is required along with imaging data.
33. Abstract: line 33 – ‘two SATB1-regulating genes, Ets2 and Bcl6’ should be changed to ‘two SATB1-regulated genes, Ets2 and Bcl6’.

Reviewer #2 (Remarks to the Author):

This manuscript reveals that SATB1 globally regulates super-enhancers of DP cells and promotes the establishment of DP cell identity. The authors provide evidence that SATB1 promotes the intra-interactions of the super-enhancers, augments the super-enhancer activity, and then enables the high expression of cell identity genes. Overall, this work provides some new information on the role of SATB1 in thymocyte development. Specific concerns are listed below. If these concerns can be adequately addressed, the reviewer supports publication in Nat. Comm..

Major concerns:

1. SATB1 can fold chromatin into loops and serves as scaffolds for SATB1-mediated tethering of regulatory regions. Meanwhile, SATB1 can assembly into oligomer through its N-terminal domain and the oligomerization plays an essential role in its binding to highly specialized DNA sequences. Here, the Hi-C data showed that interactions in super-enhancers and between super-enhancers and promoters decreased in SATB1 deficient thymocytes. Does the oligomerization of SATB1 mediate the binding in super-enhancers and between super-enhancers and promoters? It would be better if the author could design a simple experiment to verify this.
2. The authors chose two genes *Bcl6* and *Ets2* as examples to elucidate how SATB1 regulated transcription factors by super-enhancers. However, these two genes are both downregulated in *Satb1* deficient DP thymocytes. I think the authors also should choose 1-2 genes which is upregulated in *Satb1* deficient DP thymocytes and elucidate the mechanism of these genes regulated by super-enhancers.
3. Fig. 1a showed that the cells of cluster seven are significantly increased in SATB1 deletion. The another should state this phenomenon in section of results or discussion.

Minor concerns:

1. Lines 376 and 475, 100ul and 40 um should be changed to 100 μ l and 40 μ m, respectively. Similar errors exist in other text of methods. line 448, 37 C should be changed to 37 $^{\circ}$ C
2. Between number and unit should be consistently inserted a space.
3. Figure S2d, too small of the text to see clearly

Reviewer #3 (Remarks to the Author):

Delong Feng, Yanhong Chen, and colleagues have carried out a broad analysis of the genes that are regulated by SATB1 in CD4⁺ CD8⁺ DP cells, with possible mechanisms probed by mapping the association of SATB1 with loops and superenhancers in the genome of CD4⁺ CD8⁺ DP cells, and the effects of SATB1 deletion on looping and superenhancer function. They then focus on two transcription factor coding genes, *Bcl6* and *Ets2*, which appear to be strong functional targets of SATB1. The authors

show that SATB1-binding genomic regions next to these loci are vital for their expression. Altogether, this manuscript presents a large amount of high-quality, state-of-the-art genomic data for the roles and binding sites of a transcription factor that has not been fully understood before.

Still, there are some issues with the paper.

1. The tables lack legends or even Table numbers in the files themselves. These have to be provided. The reader needs to be able to see which Table is Table S1, etc. Better annotation of Table S1 is especially vital because it contains the data that establishes the biological impact of Satb1. Similarly, the figure legends throughout the paper, main figures and supplementary figures, are just skeleton legends, not adequate. At least in the supplementary figure legends, there is no space limit and these should have no reason to leave out the normal amount of useful information.

2. The effects that are shown for the SATB1 knockout seem fairly mild, as gauged by single-cell RNA-seq in Fig. 1a. However, because of the effect on Rag1, there should be a defect in positive selection. Thus, some of the genes that lose expression in SATB1-deficient cells might not be actual Satb1 targets, but simply genes that need to be upregulated by positive selection, like Zbtb7b and Runx3. I think that the authors are seeing some evidence for this difference when one compares Fig. 1d (all SATB1-ko-affected genes) with Fig. 2b and Fig. 3d (most likely, true direct SATB1 targets). If so, then the authors have the chance to make a valuable point. But there are no gene names for the heat map genes and no way to quantitate whether this is true.

a. Could the authors provide tables that list the genes in these heat maps in their order in the figure, so that the reader can understand which targets fall into which classes?

b. If it is true that the direct targets can be separated from indirect targets this way, could the authors make a statement about this?

c. It seems that the genes that could be regulated indirectly by the requirement for positive selection should have been identified before, for example, in any RNA-seq analysis of DP thymocytes from TCRalpha-deficient or MHC-deficient mice. Can the authors use data like this to check whether SATB1 roles really are confined to direct control of genes expressed before positive selection?

3. There is a discussion of the relative strengths of effects on binding to Superenhancers (SE) vs. “traditional enhancers” (isolated elements) (TE) that seems confusing in light of what is shown.

a. The paper focuses on SE and SE-proximal genes, which makes sense because these are so highly bound (Fig. S3f). But actually, in the SATB1 cKO, it seems that there is a proportionately larger effect on the number of TE (a 30% drop) than on the number of SE (a 5% drop). So even though there are detectable changes in histone modification across the SE in the absence of SATB1, the impact of SATB1 on the presence or absence of an enhancer seems to be greater for the TE. In the paragraph from lines 126 to 138, it would be good to be a little more explicit about this. If these TE changes will be ignored in

the following simply because the genes near them do not appear to change expression, then this should be stated.

b. Is there a difference between SATB1 binding to a SE, and a SATB1 Supercluster? In fact, it seems that SATB1 binding follows the H3K27ac pattern almost completely. Could the authors show an explicit comparison (e.g. Venn diagram) comparing the sites where SATB1 binds and the sites that are bound by H3K27ac? Are there any sites where SATB1 binds that are not H3K27ac sites?

4. The title of the paper claims that the main role of SATB1 is to compact super-enhancers. However, the data shown for this are very hard to evaluate quantitatively from the plots shown. Fig. 4c and f do not make it easy to see a difference between the SATB1 WT and the SATB1 cKO samples. Even the HiC data in Fig. 5a, b seem to show relatively subtle changes, compared to what one sees in other studies. (Fig. S6a is very helpful for this.)

a. Does the claim that SATB1 has a role in compaction refer to a change in loop numbers, loop length, or loop intensity? Please clarify what is meant and what is the strongest evidence for it.

b. How do the loop occurrence and loop intensity measurements come out of data like these? Fig. 4 and Fig. S5 show plots for loop number and loop length, but which data show the “loop intensity” measurement? How is the statistical confidence evaluated?

c. Fig. 4c & f and Fig. S5d need much more explanation. These data are supposed to show the “Satb1 deletion increased loop numbers but reduced the loop strength” (lines 171-172), but this is not at all obvious. What are the axes? What are the numbers on the plots? Nothing is explained in the legends to these figures.

d. Similarly, what are the dots shown in the scatter plots of Fig. 4e? Is each one an SE, and how are “loop numbers” counted for each SE? Are they raw numbers, or normalized somehow? This is not well explained either. Also, with these small changes, how great is the difference between WT and KO as compared to the difference between the two WT replicate samples, or the difference between the two KO samples?

5. Typographical errors:

a. In both Fig. 5f and Fig. S6d, the labels for Bcl6 and Ets2 data are reversed. They are both high in DP cells, but Bcl6 is low in DN3 cells, Ets2 is increasing in DN3 cells.

b. Please label Fig. 6d to show WT and KO.

c. In line 228, please insert “6” to read “the Bcl6-SE”

6. In the Discussion, line 254, the authors assert that SATB1 is a “master regulator of DP thymocytes”. Indeed, their data show a role for SATB1 in the DP cells where its activity is highest, but it is very dramatic how many DP cells are produced even without SATB1, how well their transcriptomes co-cluster with WT DP cells, and how many of them manage to go on to make SP cells despite the complete lack of

SATB1 in these lineages. This seems completely inconsistent with the normal implication of a “master regulator”. Of course this factor is worth studying, and it probably has great basic and clinical importance. But there is a huge difference between the phenotype when a paradigm-setting “master regulator” like EBF1 or PAX5 is knocked out in B cells, vs. this very subtle, modulating phenotype when SATB1 is knocked out in the T lineage.

7. The authors elegantly showed that they can completely eliminate Bcl6 and Ets2 expression in DP cells by deletion of large SE-involved regions. But the statement that Bcl6 and Ets2 at high levels “play an essential role in DP cells” (line 267) similarly seems to be in contradiction with the very mild reduction in cell number and almost completely normal CD4/CD8 profiles of DP cells that have completely lost expression of these factors in Figs. 6c,d and 7b,c.

8. Finally, this is an optional point, but I wonder if the authors have considered the chance that the SATB1 KO phenotype could be relatively weak because of another factor in the same family that may be compensating for it when it is knocked out. Are there other factors with related roles that are also expressed in these cells? Even though SATB2 is not expressed there, I wonder if some of the other CUT-domain factors could be playing a supporting role.

In summary, this paper is filled with valuable data, but the presentation right now is somewhat logically choppy, like three or four studies put together. It is too bad that in the current version, the hardest point to appreciate is the one that is stated in the title. It is hoped that the authors can clarify this.

REVIEWER COMMENTS

Reviewer #1 (Remarks to the Author):

Feng et al. investigated the mechanism of regulation of gene expression by SATB1 in DP thymocytes. Using scRNAseq analysis of SATB1-deficient thymocytes they show that the cell identity of DP thymocytes was changed, and the DP specific genes were down-regulated. They further argue that Satb1 regulates super-enhancer landscape of specific to DP cells while giving examples of 2 SATB1 target genes Bcl6 and Ets2 via deletion of their super-enhancers. Further, using HiC analysis the authors showed that interactions between super-enhancers and promoters decreased in SATB1 deficient thymocytes. This data extends on the previously known roles of SATB1 towards regulation of super-enhancers and establishment of DP cell identity.

The authors have used the power of sc-RNAseq technology to demonstrate the cell type/number changes in the thymus. Further, they show specific nature of DP signature genes' dysregulation upon Satb1 deficiency. Authors hypothesized and showed subtle changes in the global genome interactions upon Satb1 KO via HiC. The authors selected 2 of the Satb1 targets via SE analysis, namely Bcl6 and Ets2, generated KO models by deleting their super-enhancers, and demonstrated their individual roles. Although, the manuscript has good potential and approach by analyzing number of datasets, there are multiple aspects in which the authors need to strengthen their work. Many of their models and experiments require further validation. Additional experimental evidences are required for supporting multiple conclusions, e.g. to demonstrate if Satb1 actually loops the SE for Ets2 and Bcl6 to their respective promoters. The manuscript will also benefit from clear writing and elaboration of the results.

Specific Comments:

1. In Fig. 1a. the authors show that there is reduction of DN and SPs along with increase in DP cell numbers (with the example of cluster 3, line 102). Although there is a dramatic reduction in cluster 11 and a significant reduction in cluster 0, which are also DP clusters. Are there any differences in gene expression profiles for these clusters that drive their reduction compared to cluster 3? The authors should explain these contrasting populations both in the figure and the text.

Response:

Thanks for the suggestion. We analyzed differentially expressed genes between cluster 3 and cluster 0 and obtained 158 DEGs, including recombinase encoding genes *Rag1* and *Rag2*. The data was in supplementary data S1.

2. Similarly, the DN4/ISP cluster 15 has disappeared in the KO UMAP, which is contradicting their certainty as an increasing population as mentioned in the text. The authors should therefore include the corresponding flow cytometry plots for each stage to verify whether it corroborates the scRNAseq data.

Response:

Thanks for the suggestion. The effects of Satb1 deletion on thymic development have been reported previously (Alvarez JD, Yasui DH, Niida H, Joh T, Loh DY, Kohwi-Shigematsu T. *Genes Dev.* 2000 Mar 1;14(5):521-35. Kondo M, Tanaka Y, Kuwabara T, Naito T, Kohwi-Shigematsu T, Watanabe A. *J Immunol.* 2016 Jan 15;196(2):563-72.). Our results are consistent with these reports. In order for readers to better understand the results, we have added flow cytometry plots in Figure S1.

3. Additionally, the B cell number is dramatically increased in the KO condition as shown in cluster 7. Is there a B cell specific gene downregulation associated with Satb1 KO?

Response:

It is an interesting question. Actually we are also interested in the increased B cell number in Satb1 deficient thymus and initiated a new project on it. But we didn't see much downregulation of the B cell feature genes in the KO thymus.

4. How correlative is the scRNAseq data from DP bulk RNA-seq data? The authors should discuss this as the gene expression profiles are not always similar in scRNAseq data. Further validation would be required.

Response:

Thanks for the nice suggestion. We compared the expression profiles and they are highly correlative ($r=0.79$) (Fig. S2d). However, single cell RNA-seq did not perform well in the detection of low- and medium-expressed genes, and the differential genes obtained from scRNA-seq were less than bulk RNA-seq. We also validated expression of some genes using quantitative PCR (Fig. S2e).

The description in page 5

"DEGs from the bulk RNA-seq is highly correlated to that of scRNA-seq ($r=0.79$, Fig. S2d)."

5. The text (line 106) does not correlate with the associated figure S2, as there is opposite trend shown for up-and-down genes. The authors should carefully check for such inconsistencies. Further, for S2e it seems unclear what the authors wish to conclude from the same.

Response:

Thanks. It is fixed.

“There were 928 downregulated genes and 576 upregulated genes in *Satb1* deficient DP thymocytes (Fig. S2b-c and Supplementary data S1).”

In the figure S2e, we showed the percentage of the up-regulated genes or down-regulated genes with expression peak during thymocyte development. For example, 27% up-regulated genes reach the expression peak in the DN1 stage and only 10.1% up-regulated genes reach the expression peaks in the DP stage. On the contrary, only 8.6% down-regulated reach the expression peak in the DN1 stage and 36.8% down-regulated gene reach their expression peak in the DP stage.

The description in Page 5

“We noticed that some of the upregulated genes in *Satb1* deficient DP cells are the feature genes in the early stages of thymocyte development, such as the *Il2ra* gene encoding CD25, a cell surface maker of DN2/3 cells (Fig. 1c and S1c). We also confirmed some upregulated genes by quantitative PCR (Fig. S2e). To learn expression characteristics of the upregulated genes during thymocyte development, we analyzed expression data of nine developmental stages (from DN1 to SP) obtained from ImmGen Datasets³⁴. Most of the upregulated genes are highly expressed in thymocyte development earlier stages like DN1 and DN2a (Fig. 1d, 1e and S2f). About 28% of the upregulated genes have an expression peak in the DN1 stage, while 10% of these genes have a peak in DP cells (Fig. S2f). 77% of upregulated genes have an expression peak in earlier stages (from DN1 to ISP).”

6. In Fig. 1d, the authors posit that downregulated genes are mostly from the DP stage, whereas the upregulated genes belong to the prior stages. It seems that they have generated the heatmap from the scRNA expression data from all stages, but it is unclear both in the text and legend. If it is the bulk RNA data, then this should be explained. Similarly, the results from many other figures are not clearly elaborated. The authors should explain the figures clearly throughout the manuscript.

Response:

Thanks for the suggestion. The expression data of all stages is from ImmGen Datasets. We provided all data information in the results of the new version.

The description in Page 5

“We noticed that some of the upregulated genes in *Satb1* deficient DP cells are the feature genes in the early stages of thymocyte development, such as the *Il2ra* gene encoding CD25, a cell surface maker of DN2/3 cells (Fig. 1c and S1c). To learn expression characteristics of the upregulated genes during thymocyte development, we analyzed expression data of nine stages obtained from ImmGen Datasets³⁴.”

7. The authors further claim that DN specific genes are upregulated in DPs, but the figure indicates that a similar ratio of downregulated genes is also present for the DN stages as well. How do the authors reconcile these findings?

Response:

We analyzed the expression peak stages of all DEGs and exhibited it in Fig. S2f. We can see 27% of the upregulated genes have an expression peak in the DN1 stage, while 9% of down-regulated genes have a peak in the DN1 stage. We accumulated expression peak percentages of the earlier stages (from DN1 to ISP) and found that 77% of upregulated genes in Satb1cKO DP have an expression peak in earlier stages, it is 22.7% for downregulated genes. To describe the result more accurately, we have rewritten the corresponding content.

The description in page 5

“We noticed that some of the upregulated genes in Satb1 deficient DP cells are the feature genes in the early stages of thymocyte development, such as the *Ii2ra* gene encoding CD25, a cell surface marker of DN2/3 cells (Fig. 1c and S1c). We also confirmed some upregulated genes by quantitative PCR (Fig. S2e). To learn expression characteristics of the upregulated genes during thymocyte development, we analyzed expression data of nine developmental stages (from DN1 to SP) obtained from ImmGen Datasets³⁴. Most of the upregulated genes are highly expressed in thymocyte development earlier stages like DN1 and DN2a (Fig. 1d, 1e and S2f). About 28% of the upregulated genes have an expression peak in the DN1 stage, while 10% of these genes have a peak in DP cells (Fig. S2f). 77% of upregulated genes have an expression peak in earlier stages (from DN1 to ISP).”

8. Fig. 1f: The authors state that Satb1 mediated repression is tested using GSEA with DN1/2 gene sets. It is not clear in the figure/text which genes they have used as their ranked list for the statistic. It is all the more required since using up- or down- regulated genes would change the interpretation of the leading edge of the plot, hence the conclusion. These findings should be validated by quantifying the gene expression using PCR assays.

Response:

Thanks for the suggestion. We revised it and the genesets were showed in Supplementary data S2. We also used qPCR to validate the gene expression of some upregulated genes, which are shown in figure S2e.

9. Fig. 2, the super-enhancer landscape is thought to be established a priori the stage in which the gene expression is controlled. The authors should also monitor the super-enhancers in DN4/ISP stages and how the expression of those genes are affected in DP. Importantly, as many of the super-enhancer gene expression show increased correlation with DP super-enhancers (shown in 2b) in previous stages, the authors should experimentally validate these bindings and gene expression in stage-wise manner. The second sub-panel of 2b should be shifted to supplementary data.

Response:

Thanks for the nice suggestion. We did the analysis and the result showed that super-enhancer regions are accessible in the ISP stage (Fig. S3a). The sub-panel of 2b was moved to Fig. S3. We added the result in figure S3a and the description in Page 6. We experienced a technical difficulty in validating histone modifications and SATB1 bindings in a stage-wise manner because of the limitation of cell numbers of earlier stages.

"We analyzed chromatin accessibility of DP super-enhancer regions in each stage of thymocyte development using ATAC-seq data from ImmGen Datasets³⁴. Accessibility of super-enhancer regions increased from DN1 and peaked at ISP and dropped in DP (Fig. S3a), indicating establishment of DP super-enhancers in the ISP stage."

10. In Fig. 2c, are the ChIP-seq datasets used derived from sorted DPs? Its not clear in either the figure or the text. The authors should revise their text to be more elaborative since a single stage is in question and the conclusions are drawn for specificity of Satb1 function in DPs. Furthermore, the authors should include experimental data for the mined data to ascertain its correlation with their own datasets.

Response:

Thanks for the suggestion. All ChIP-seq data used here are derived from sorted DP cells. The data are previously published, except the SATB1 ChIP-seq in Satb1 deficient cells. The information is in the data availability section in Page 23.

"The raw sequence data of single-cell RNA-Seq, bulk RNA-seq, ChIP-seq, Hi-C, 4C-seq, 3C-HTGTS, and 5'RACE reported in this paper have been deposited in the Gene Expression Omnibus (GEO) database under the accession number: GSE182995.

GEO accession codes (or SRA accession number) of the published data used in this study are as follows: H3K4me3 ChIP-seq of CD4⁺ CD8⁺ DP thymocytes, GSE21207; Rad21 and Nipbl ChIP-seq, GSE48763; CTCF ChIP-seq, GSE141223; Satb1 ChIP-seq, GSE90635; RNA-seq of T cell development, GSE109125; H3K27ac ChIP-seq of Satb1WT and Satb1cKO DP thymocytes, DRP003376; H3K4me1 ChIP-seq of Satb1WT and Satb1cKO DP thymocytes, DRP003376; 3e Hi-C data, GSE79422."

11. In the context of Fig. 2d, the authors show a modest enrichment of Satb1 at the K27ac enriched regions, however its binding seems to be opposite to the modification when they plotted the average plot around Satb1 peaks. The authors should discuss this in details as the conclusions drawn are

affected. Authors also mention that there is promoter binding, but data is presented only as distance from super-enhancers and overall promoter binding (S3e), hence warrants inclusion of supporting data in form of distance tag plots.

Response:

Thanks for pointing it out.

In fact, it was reported that SATB1 had opposite effects on histone acetylation by recruiting the histone deacetylase HDAC1 or acetylase CBP/p300. SATB1 occupancy had the same direction in cluster3, while cluster 4 displayed a strong opposite direction between SATB1 and H3K27ac (Fig. S3f). It is consistent with the SATB1's opposite functions in regulating histone acetylation. However, SATB1 promoted histone acetylation in super-enhancer regions, which can be concluded from the observation that Satb1 deletion reduced H3K27 acetylation of SEs (Fig. 2e).

“To learn the relationship between SATB1 and super-enhancers, we reanalyzed the previous SATB1 ChIP-seq data with sorted DP thymocytes²⁸. It was reported that active promoters can produce false-positive peaks in ChIP-seq experiments³⁷. We did SATB1 ChIP-seq in Satb1 deficient thymocytes from Satb1^{fl/fl} × CD4-cre mice. It showed that the SATB1 binding in active regions was specific in SATB1-expressing WT cells (Fig. 2c). We noticed that many super-enhancers had high SATB1 occupancy, like the Cd4 locus (Fig. 2c). Then we did an overlapping analysis of SATB1 peaks with published ChIP-seq data of the active histone modification markers histone H3K4 monomethylation (H3K4me1)²⁸, H3K4me3³⁸, H3K27ac²⁸, and chromatin organization complex Rad21, Nipbl³⁹, and CTCF⁴⁰. SATB1 binding sites have high RAD21, CTCF, and Nipbl occupancy, especially for cluster 1 and 3 (Fig. S3c and d). SATB1 occupancy overlapped with H3K4me3 in many sites and displayed the same direction in the average plot (Fig. S3e and f). While SATB1 binding displayed an opposite direction with H3K4me1 and H3K27ac on average (Fig. S3e). We noticed that SATB1 occupancy had the same direction in cluster3, while cluster 4 displayed a strong opposite direction between SATB1 and H3K27ac (Fig. S3f). It was reported that SATB1 had opposite effects on histone acetylation by recruiting the histone deacetylase HDAC1 or acetylase CBP/p300⁴¹, which may explain the two conditions of H3K27ac modification on SATB1 occupied sites.” in Page 6

12. In Fig. 2e, the authors show the modest decrease in K27ac occupancy plotted for SEs. Although there is a much dramatic decrease in TEs (shown in supplementary). The authors should discuss this important distinction in the text. Additionally, the authors should show the percentage of gained and lost SEs upon Satb1 KO.

Response:

Thanks for the suggestion. The percentages of gained and lost SEs have added in the Fig. 2f.

“The SATB1 signals were enriched in super-enhancers (Fig. 2d and S3g-h). About 95% of super-enhancers have SATB1 occupancy, while it is only 20% for traditional enhancers (Fig. S3h). We observed a correlation between SATB1 occupancy and H3K27ac modification in super-enhancer regions (Fig. 2d), indicating that SATB1 may recruit histone acetylases instead of deacetylases to super-enhancers. Consistent with it, SATB1 deficiency leads to a significant decrease of H3K27 acetylation in super-enhancer regions (Fig. 2e). The average intensity of H3K4me1 in super-enhancers was reduced in SATB1 deficient cells (Fig. S3j). SATB1 deficiency changed the landscape of super-enhancers in DP thymocytes, 118 super-enhancers were lost, and 106 new super-enhancers were gained (Fig. 2f). The result showed that SATB1 regulated activity of super-enhancers in DP thymocytes.” in Page 6

13. In Fig. 2g, the authors posit that more genes are downregulated associated with SEs upon Satb1 KO. According to the figure, the authors seem to have used all SEs and not just the ~246 SEs they identified specific to DP in Fig. 2a. The authors should revise all the plots wherein they show a DP specificity to bolster their findings. 13-2. Are the gained SE and gene expression (2h) not 'DP specific' genes?

Response:

The Fig. 2g and 2h used the 246 DP-SEs we identified in Fig. 2a and the list is in Supplementary data S3. In Fig.2g , relative expression of 246 SE-associated genes are shown in WT and Satb1cKO DP cells.

We identified super-enhancers by enhancers ranks with ROSE. The cutoff value of H3K27ac signal are difference between WT(H3K27ac signal=16941) and KO(H3K27ac signal=14437) groups. We find that over 90% gained SE are overlapped with TE regions of WT DPs, which means the gained SEs come from TEs of WT DP (Fig. S3I). The gained SE-associated genes contains DN or DP specific genes.

14. Fig. 2o, and related text in line 316 does not correlate with the supplementary data which showed that TEs are more affected than SEs in Satb1 KO. The authors should reconcile these findings.

Response:

The TE number reduced in Satb1cKO cells (Fig. S3i). However, SE-associated genes are more sensitive to SATB1 deficiency than traditional-enhancer-associated genes (Fig. 2j). We revised it in the new version.

“Then we analyzed the expression of SE-associated genes in SATB1 deficient cells. The GSEA analysis showed that all SE-associated and lost-SE-associated genes are significantly enriched in downregulated genes of SATB1 deficient cells (Fig. 2g). 67% of SE-associated genes were downregulated and 33% upregulated (Fig. 2h). The expression reduction of lost-SE-associated genes was more significant, and the maintained-SE-associated genes were downregulated (Fig. 2i). SE-associated genes are more sensitive to SATB1 deficiency than traditional-enhancer-associated genes (Fig. 2j). The gained-SE-associated genes were enriched in the upregulated gene of SATB1 deficient cells (Fig. 2i and S3). The gained-SEs were generally transferred from traditional enhancers and most gained-SE-associated genes have an expression peak in the DP stage (Fig. S3l and 3m), indicating that these genes didn’t represent the earlier stage genes repressed by SATB1. Taken from all the above results, it was suggested that SATB1 regulated the DP signature genes by activating super-enhancers.”

15. Fig. 3b. What is the dataset used to rank the genes? The authors also need to show the association of SEs identified with Satb1 enriched cluster plot.

Response:

Thanks for the suggestion.

We used an intuitive way to show the relationship between Satb1 clusters and H3K27ac in the new version (Fig. 3c). The Satb1 SC regions are from Fig. 3a. The H3K27ac dataset is same as in the Figure 2. The overlap of SEs and SATB1 SCs is shown in Fig. 3b.

16. In Fig 3d, as the plotting is done around DP for highly dysregulated genes, the authors should also plot the heatmaps around DN4/ISP, so as to strengthen the conclusion of DP specificity of expression. Further, this plot should be moved to Fig. 1, along with the overall scRNAseq data.

Response:

The data in Fig. 3d (Fig. 3e in the new version) is used to show the DP special expression of the SC-associated genes and the role of SATB1 in regulating genes of the DP identity. The expression data are derived from bulk RNA-seq experiments in ImmGen datasets instead of the scRNA-seq data. So we think it would be better in Figure 3. We revised this part in the new version to avoid any ambiguity.

“We analyzed the expression of the SATB1 clusters-associated genes during thymocyte development using the gene expression data from ImmGen Datasets. SATB1 SC-associated genes are highly expressed in DP thymocytes, and many of them are specifically expressed in the DP stage (Fig. 3e).”

17. Figure 3e. Since the authors posit that there is strong association of Satb1 with SEs of DP, the pathways very strongly enriched for Satb1 in DPs are only enriched with much lower confidence and genes in DP SEs. Can the authors also show processes more highly enriched for high confidence SE genes.

Response:

Confidence in the GO analysis is related to the geneset size. There are 246 SE-associated genes and 603 Satb1SC-associated genes, which is the main factor of generating the difference of enrichment and FDR values. We show Top 5 GO pathways in each groups, and 3 pathways are overlapped, so there are Top 7 GO pathway are shown in Fig.3e.

18. In Fig. 4 b, the authors plotted the densities of Satb, Ctf and K27ac for regions of lost and gained loops in the KO condition. Although it is clear that Satb1 and Ctf occupancy is more on the lost interactions, K27ac is only modestly more, which suggests its ubiquitous binding irrespective of Satb1. The authors should present detailed plots to reveal the minor differences, and the actual concordance between the lost and gained analysis among the three.

Response:

Thanks for the suggestion. We made a H3K27ac heatmap for the regions of gained and lost loops and the difference is much clear. We added the heatmap in fig. S5d.

19. Fig. 4c, the authors should also plot the counts of loops in the KO condition. Its mentioned in the text but not plotted per se. Additionally the supplementary figures the authors referred to exhibit much more dramatic differences (assuming prep 1 and prep2). Since the difference in the two preps shown is quite high, the authors should discuss this in the text.

Response:

Thanks for the suggestion. We carefully reviewed the analysis process and found an inconsistency that we used all loops from WT and Satb1cKO for the pile-up plots in the old version (Fig. 4c). We reanalyzed the data and optimized the color contrast. Now we only used the loops obtained from the wild-type Hi-C data for analysis. It can be seen that the knockout of SATB1 has a great impact on the loops, and the two sets of data are consistent.

20. The authors mention in line 173, 'The loop strength increased in the promoters of the upregulated genes and decreased in the downregulated genes'. Are the increased loop strength genes of DN lineage? These analyses will further strengthen the major point of the manuscript.

Response:

Thanks for the nice suggestion. We analyzed the loops in the DN lineage genes and found that few genes of DN lineage have chromatin loops at promoters. There are two possibilities: the one is that the DN specific genes are still in low expression although SATB1 deletion upregulated these genes; the second is that chromatin loops are unnecessary for expression of the DN specific genes.

21. Fig. 5, experimental evidence showing if Satb1 actually loops the SE for *Ets2* and *Bcl6* to their respective promoters is required for the conclusions drawn.

Response:

Actually we confirmed the Hi-C result using 3C-HTGTS assay, a 4C-like technique developed by Frederick Alt group (Jain S. et al. Cell 2018 174(1):102-116). We found the technique generating much better signals than 4C-seq and did it instead of 4C in this study. We added a description in the text. The interactions between promoters and SEs were significantly reduced in Satb1 deficient DP thymocytes based on 3C-HTGTS data.

“To confirm enhancer-promoter interactions of the two genes, we performed 3C-HTGTS assay⁵⁵, a new technique like the Circular Chromosome Conformation Capture (4C). We prepared a 3C library from sorted DP cells and did 3C-HTGTS assay with a bait for the *Ets2* or *Bcl6* promoter, respectively.”

22. Fig. 5b and c, the 3C-HTGTS data shows much broader interaction extending on either sides from the SE (assuming its labeled red), which also decreases. The authors should discuss this in the text. This will also support the next Fig. 6b.

Response:

Thanks. We discussed it in the text.

“To confirm enhancer-promoter interactions of the two genes, we performed 3C-HTGTS assay⁵⁵, a new technique like the Circular Chromosome Conformation Capture (4C). We prepared a 3C library from sorted DP cells and did 3C-HTGTS assay with a bait for the *Ets2* or *Bcl6* promoter, respectively. 3C-HTGTS data revealed that the *Bcl6* promoter had broad interactions with around 1Mb upstream region and interactions were dense in the SE region (Fig. 5c). The interactions with the SE region reduced dramatically in SATB1 deficient thymocytes, while the interactions with the distal CTCF binding sites remained a same level (Fig. 5c). Interactions of the *Ets2* promoter were concentrated in the downstream SE region and interactions reduced substantially in SATB1 deficient cells (Fig. 5d). The H3K27 acetylation of the super-enhancers and promoters

also reduced dramatically (Fig. 5c and 5d), indicating reduced chromatin interactions disturbed activities of super-enhancers. ”

23. The authors further mention that ETS SE -/- results in reduced cell number by 69% (line 216) whereas Fig 6c shows at modest ~30%. These need to be confirmed.

Response:

Thanks for pointing it out. It was fixed in the text.

24. According to Fig. 6d, the DN/DP ratios do not correspond with the Satb1 KO ratios. The authors should discuss this, as they claim Satb1 mediated effector function by Ets2.

Response:

Thanks. We don’t think that SATB1 mediated functions in DP thymocytes by ETS2. STAB1 regulates many functional characteristics of DP cells by affecting the expression of key genes in DP cells, including the rearrangement of *Tcra* by regulating Rag1 and Rag2 genes. Previous reports (Fisher IB. et al. JBC 2012 287(8):5199-210. and Zaldumbide, A. et al. JI 2002 169:4873-4881) showed that mice with dominant negative Ets2 mutants displayed reduced thymus size associated with a 60-80% reduction in thymocyte populations, which is similar to the observation in Est2-SE deleted mice. However, Ets2 isn’t involved in the regulation of recombinase Rag1 and Rag2 and Tcra rearrangement is normal in Ets2-SE deleted mice. The deficient Tcra rearrangement blocks DP development and increased DP cell percentage, which may explain the difference of the DN/DP rations between Satb1 KO and Ets2-SE KO mice.

25. Fig. 6d: the conditions for FACS plots are not mentioned. Same issues also applicable to Fig. 7.
Response:

Thanks for pointing it out. It was fixed in the new version.

26. In case of ETS2 or BCL6 SE KO, is there any change in Satb1 targets? Since it seems that sections covering Fig.s 6 and 7 are separate results in themselves, the authors should perform RNA sequencing or at least qPCRs in these conditions as well. Is there any phenotypic/cellular characteristics that match with Satb1?
Response:

Thanks for the suggestion. SATB1 regulates multiple facets of DP characteristics, and Ets2 and Bcl6 only regulate one of them. As our response to the comment 24, the phenotype of Ets2-SE KO was covered in SATB1 deleted mice by other defectives. But we did see the deficient Tcra rearrangement in Bcl6-SE mice, which is also observed in SATB1 deficient thymocytes. We are conducting follow-up studies and hope to present the data in the next report.

27. In Figs 6 and 7, the authors show that expression of ETS2 and BCL6 decrease upon their respective SE KO. These findings should be validated further by providing protein data for KO and WT mice.

Response:

Thanks for the suggestion. Western blot results was added in Fig. 6 and 7, respectively.

28. Since the authors generated specific mice lines, it is of utmost importance to show validate their KO models via the confirmation of deletion of genomic regions, as well as phenotypic characterization such as organ size etc.

Response:

Thanks for the suggestion. The genome deletions were confirmed by Sanger sequencing and the chromatogram plots showing deletion-rejoined sites were added in Fig. S7. We didn't observed any visible phenotypic abnormalities of these two lines.

Ets2-SE KO

Bcl6-SE KO

29. Since Ets1 KO is reported previously (Eyquem et al 2004) to exhibit reduced DP and SPs as DN population increases. However, the transgenic line used here does not recapitulate the same to the full extent especially at SP level (Fig 6.) Hence, the authors need to show the differences and discuss the same in the discussion.

Response:

Thanks for the suggestion. Eyquem et al reported that Ets1^{-/-} displayed impaired DN3-to-DP transition due to death of DN4 cells. They also observed reduced cell numbers of DP and SPs. However, two reports with dominant-negative Ets2 mutant transgenic mice showed that Ets2 had few effect on SP cells. We speculated that the difference in the effects of Ets1 and Ets2 on thymocyte development might be due to the persistent high expression of Ets1 in SP cells, while the expression of Ets2 was down-regulated in SP cells. The discussion was added.

“Ets1 and Ets2 are members of the *ets* family of transcription factors and play a role in thymocyte development, especial for DN-to-DP transition. Ets1^{-/-} mice displayed impaired DN3-to-DP transition due to susceptible to cell death of ND4⁶⁴. The research on the dominant-negative truncated Ets2 transgenic mice and a phosphomutant Ets2 (T72A) transgenic mice showed that Ets2 plays an essential role of Ets2 in thymocyte development^{51,52}. In this study, Ets2-SE mice have a similar phenotype, indicating that Ets2 high expression regulated by the super-enhancer is critical for the development and survival of DP cells. Although Ets1 is also regulated by a super-enhancer in DP thymocytes, SATB deletion did not affect its expression, suggesting that other chromatin organizing proteins may be involved in its regulation.”

30. According to the data presented, the Bcl6 SE KO shows no change in thymic populations as well as recombination. Since the authors claim no role of Bcl6, they should confirm by recapitulating the findings (as supplementary) of the Bcl6^{-/-} study which showed its role in TH2/Tfh differentiation.

Response:

Thanks for the suggestion. We did see that Bcl6-SE KO mice displayed a slight distortion of Tcra rearrangement (Fig. 7g and 7h), which was weaker than the phenotype of Satb1-null mice. Actually,

we initiated a new project on Bcl6 and Tcra rearrangement. We did check Tfh differentiation of Bcl6-SE KO mice because Bcl6 plays a critical role in germinal center. We have some interesting findings. However, we feel that these results are too far from the subject of this study and therefore did not put them in this paper.

31. It is essential to perform Satb1 looping validation for the Ets1 and Bcl6 loci in WT vs KO condition. This will highlight one of the major new findings of this manuscript, showing the role of SATB1 in the regulation of their SE.

Response:

Thanks. We validated enhancer-promoter interactions of Ets2 and Bcl6 using 4C or 3C-HTGTS in WT and KO, which was exhibited in Fig. 6b and 7b. Ets2 and Bcl6 promoters have a strong interactions with super-enhancer. SEs deletion dramatically changed chromatin organization in loci.

32. Validation experiments such as WB, imaging etc. are much needed here, since both Fig.s 6 and 7 depict SE deletion, which may or may not change its expression in a cell type/Satb1-dependent manner. Hence, WB of sorted populations is required along with imaging data.

Response:

Thanks for the suggestion. The western blot results were added in figures (Fig. 5f, 6c, and 7c).

33. Abstract: line 33 – ‘two SATB1-regulating genes, Ets2 and Bcl6’ should be changed to ‘two SATB1-regulated genes, Ets2 and Bcl6’.

Response:

Thanks. It was fixed.

Reviewer #2 (Remarks to the Author):

This manuscript reveals that SATB1 globally regulates super-enhancers of DP cells and promotes the establishment of DP cell identity. The authors provide evidence that SATB1 promotes the intra-interactions of the super-enhancers, augments the super-enhancer activity, and then enables the high expression of cell identity genes. Overall, this work provides some new information on the role of SATB1 in thymocyte development. Specific concerns are listed below. If these concerns can be adequately addressed, the reviewer supports publication in Nat. Comm..

Major concerns:

1. SATB1 can fold chromatin into loops and serves as scaffolds for SATB1-mediated tethering of regulatory regions. Meanwhile, SATB1 can assembly into oligomer through its N-terminal domain and the oligomerization plays an essential role in its binding to highly specialized DNA sequences. Here, the Hi-C data showed that interactions in super-enhancers and between super-enhancers and promoters decreased in SATB1 deficient thymocytes. Does the oligomerization of SATB1 mediate the binding in super-enhancers and between super-enhancers and promoters? It would be better if the author could design a simple experiment to verify this.

Response:

Thanks for the suggestion. It is a very interesting question how the oligomerization of SATB1 mediates chromatin interactions. Actually it was reported that disruption of STAB1 oligomerization changed expression of over 1000 genes (Zheng, M. et al. Molecular and Cellular Biochemistry, 430:171–178 (2017)). It was also proposed that the oligomerization of SATB1 may be related to liquid-liquid phase separation (Zelenka T. and Spilianakis C. Nucleus 11:1, 117-131 (2020) and Zelenka T. et al. Biorxiv (2021)). So it is a complicate question and we don't think that a simple experiment can answer it.

2. The authors chose two genes Bcl6 and Ets2 as examples to elucidate how SATB1 regulated transcription factors by super-enhancers. However, these two genes are both downregulated in Satb1 deficient DP thymocytes. I think the authors also should choose 1-2 genes which is upregulated in Satb1 deficient DP thymocytes and elucidate the mechanism of these genes regulated by super-enhancers.

Response:

Thanks for the suggestion. We validated several up-regulated genes in Satb1 deficient cells and added in Fig. S2e. It was reported that SATB1 had opposite effects on histone acetylation by recruiting the histone deacetylase HDAC1 or acetylase CBP/p300. Most of up-regulated genes are expressed in earlier stages, and Satb1 may repress these genes by recruiting HDAC directly or via

other factors indirectly. But we don't think that SATB1 plays repressive function via super-enhancers because our data showed that H3K27ac levels of SEs in DP cells were reduced in SATB1 deficient cells. We are more interested in genes specifically expressed in DP thymocytes because they play an essential role in the identity of DP thymocytes.

3. Fig. 1a showed that the cells of cluster seven are significantly increased in SATB1 deletion. The other should state this phenomenon in section of results or discussion.

Response:

Thanks for the suggestion. It is an interesting phenomenon. Actually we initiated a new project on it. We added a description in results.

“The B cell number also increased in the Satb1 deficient thymus (Fig. 1a and S1a). A recent report showed that Satb1 plays a role in B cell survival and maturation³². The increased B cell number may be an intrinsic feature due to the Satb1 deletion in hematopoietic stem cells.”

Minor concerns:

1. Lines 376 and 475, 100ul and 40 um should be changed to 100 μ l and 40 μ m, respectively. Similar errors exist in other text of methods. line 448, 37 C should be changed to 37 $^{\circ}$ C
2. Between number and unit should be consistently inserted a space.
3. Figure S2d, too small of the text to see clearly

Response:

Thanks a lot. All of them are fixed

Reviewer #3 (Remarks to the Author):

Delong Feng, Yanhong Chen, and colleagues have carried out a broad analysis of the genes that are regulated by SATB1 in CD4+ CD8+ DP cells, with possible mechanisms probed by mapping the association of SATB1 with loops and superenhancers in the genome of CD4+ CD8+ DP cells, and the effects of SATB1 deletion on looping and superenhancer function. They then focus on two transcription factor coding genes, Bcl6 and Ets2, which appear to be strong functional targets of SATB1. The authors show that SATB1-binding genomic regions next to these loci are vital for their expression. Altogether, this manuscript presents a large amount of high-quality, state-of-the-art genomic data for the roles and binding sites of a transcription factor that has not been fully understood before.

Still, there are some issues with the paper.

1. The tables lack legends or even Table numbers in the files themselves. These have to be

provided. The reader needs to be able to see which Table is Table S1, etc. Better annotation of Table S1 is especially vital because it contains the data that establishes the biological impact of Satb1. Similarly, the figure legends throughout the paper, main figures and supplementary figures, are just **skeleton legends, not adequate**. At least in the supplementary figure legends, there is no space limit and these should have no reason to leave out the normal amount of useful information.

Response:

Thanks for the suggestion. We added more annotation and information in tables and figure legends.

2. The effects that are shown for the SATB1 knockout seem fairly mild, as gauged by single-cell RNA-seq in Fig. 1a. However, because of the effect on Rag1, there should be a defect in positive selection. Thus, some of the genes that lose expression in SATB1-deficient cells might not be actual Satb1 targets, but simply genes that need to be upregulated by positive selection, like Zbtb7b and Runx3. I think that the authors are seeing some evidence for this difference when one compares Fig. 1d (all SATB1-ko-affected genes) with Fig. 2b and Fig. 3d (most likely, true direct SATB1 targets). If so, then the authors have the chance to make a valuable point. But there are no gene names for the heat map genes and no way to quantitate whether this is true.

Response:

Thanks for the suggestion. Satb1 plays versatile roles in thymocyte development. Actually, Satb1 is involved in positive and negative selection (Kondo, M. et al. J Immunol (2016) 196(2):563-72) and regulates lineage-specifying factors, including ThPOK(Zbtb7b), Runx3, CD4, CD8, and Treg factor Foxp3, *via* regulating enhancers in these genes in a locus-specific manner (Kakugawa, K. et al. Cell Reports (2017) 19, 1176–1188). We agree that many DEGs are not direct targets of SATB1 and provided data related to heatmaps in Supplementary data S6.

a. Could the authors provide tables that list the genes in these heat maps in their order in the figure, so that the reader can understand which targets fall into which classes?

Response:

We provided heatmap data in Supplementary data S6. Actually, we will provide all data related to figures. It is the policy of the journal.

b. If it is true that the direct targets can be separated from indirect targets this way, could the authors make a statement about this?

Response:

Actually, it has been reported that SATB1 directly regulated Zbtb7b and Runx3 (Kakugawa, K. et al. Cell Reports (2017) 19, 1176–1188). Generally, SATB1 binding on the regulatory elements of the target genes are considered evidence of direct regulation. Most of super-enhancers (222/234) in DP thymocytes have SATB1 occupancy.

c. It seems that the genes that could be regulated indirectly by the requirement for positive selection should have been identified before, for example, in any RNA-seq analysis of DP thymocytes from TCRalpha-deficient or MHC-deficient mice. Can the authors use data like this to check whether SATB1 roles really are confined to direct control of genes expressed before positive selection?

Response:

It is a nice suggestion. In fact, the proportion of post-selection DP cells is very small, generally CD69⁺ cells only account for 5% of DP cells. In this way, the RNA-seq performed on our sorted DP cells was actually mostly representative of the cells before selection. The major biology

process in DP cells before selection is Tcr α rearrangement. We reported previously that SATB1 regulated Tcr α rearrangement via regulating recombinase Rag1 expression.

3. There is a discussion of the relative strengths of effects on binding to Superenhancers (SE) vs. “traditional enhancers” (isolated elements) (TE) that seems confusing in light of what is shown.
- a. The paper focuses on SE and SE-proximal genes, which makes sense because these are so highly bound (Fig. S3f). But actually, in the SATB1 cKO, it seems that there is a proportionately larger effect on the number of TE (a 30% drop) than on the number of SE (a 5% drop). So even though there are detectable changes in histone modification across the SE in the absence of SATB1, the impact of SATB1 on the presence or absence of an enhancer seems to be greater for the TE. In the paragraph from lines 126 to 138, it would be good to be a little more explicit about this. If these TE changes will be ignored in the following simply because the genes near them do not appear to change expression, then this should be stated.

Response:

Intuitively, the reduction in the number of SEs in SATB1cKO DP cells is less obvious than the reduction in the number of TEs. However, about half of the WT SEs were actually lost (118/246) in SATB1cKO DP cells, while the newly gained SEs were mostly converted from TEs (Fig. 2f and S3l). This suggests that the absence of SATB1 largely affects SE composition. On the other hand, according to the ROSE algorithm, the cutoff taken by the SE of SATB1cKO is different from that of WT and is actually lower than that of wild type, which actually makes some weak signal regions SE. We added a statement in the results.

“The SATB1 signals were enriched in super-enhancers (Fig. 2d and S3g-h). About 95% of super-enhancers have SATB1 occupancy, while it is only 20% for traditional enhancers (Fig. S3h). We observed a correlation between SATB1 occupancy and H3K27ac modification in super-enhancer regions (Fig. 2d), indicating that SATB1 may recruit histone acetylases instead of deacetylases to super-enhancers. Consistent with it, SATB1 deficiency leads to a significant decrease of H3K27 acetylation in super-enhancer regions (Fig. 2e). The average intensity of H3K4me1 in super-enhancers was reduced in SATB1 deficient cells (Fig. S3j). SATB1 deficiency changed the landscape of super-enhancers in DP thymocytes, 118 super-enhancers were lost, and 106 new super-enhancers gained (Fig. 2f). The result showed that SATB1 regulated activity of super-enhancers in DP thymocytes.”

- b. Is there a difference between SATB1 binding to a SE, and a SATB1 Supercluster? In fact, it seems that SATB1 binding follows the H3K27ac pattern almost completely. Could the authors show an explicit comparison (e.g. Venn diagram) comparing the sites where SATB1 binds and the sites that are bound by H3K27ac? Are there any sites where SATB1 binds that are not H3K27ac sites?

Response:

Thanks for the suggestion.

The Venn diagram showed that 63.8% (157/246) of SEs were overlapped with SATB1 SCs (Fig. 3b). 12 of 603 SATB1 SCs didn't have H3K27ac peaks, while 42.6% of Satb1 peaks didn't have H3K27ac. It was reported that SATB1 had opposite effects on histone acetylation by recruiting the histone deacetylase HDAC1 or acetylase CBP/p300. We also observed a strong opposite direction between SATB1 and H3K27ac in cluster 4 (Fig. S3f).

4. The title of the paper claims that the main role of SATB1 is to compact super-enhancers. However, the data shown for this are very hard to evaluate quantitatively from the plots shown. Fig. 4c and f do not make it easy to see a difference between the SATB1 WT and the SATB1 cKO samples. Even the HiC data in Fig. 5a, b seem to show relatively subtle changes, compared to what one sees in other studies. (Fig. S6a is very helpful for this.)

a. Does the claim that SATB1 has a role in compaction refer to a change in loop numbers, loop length, or loop intensity? Please clarify what is meant and what is the strongest evidence for it.

Response:

Chromatin loops here are statistically confident chromatin contacts, which called by a bioinformatic tool Fit-Hi-C (Ferhat Ay, Timothy L. Bailey, William S. Noble. 2014. *Genome Research*. 24(6):999-1011, 2014.). We did pile-up analysis to show average changes of all loops, enhancer-promoter loops, and SE-associated loops in *Satb1* deficient DP thymocytes using a software coolpup.py (Flyamer IM, Illingworth RS, Bickmore WA. *Bioinformatics*. 2020 May 1;36(10):2980-2985.).

We carefully reviewed the analysis process and found an inconsistency that we used all loops from WT and *Satb1*cKO for the pile-up plots in the old version (Fig. 4c). We reanalyzed the data and optimized the color contrast. Now we only used the loops from the wild-type Hi-C data for analysis. It can be seen that the knockout of SATB1 has a great impact on the loops identified in WT cells (The numbers in the plots represent the value of the highest normalized contacts), and the two sets of data are more consistent (Fig. 4c and Fig. S5e).

We also noted that these loop sizes were significantly reduced. We believe that the reduction in loop sizes reflect an overall loosening of chromatin (Fig. 4d). This may be even more important for SE, as recent studies have shown that multiple enhancers in SEs cluster together, exhibiting the characteristics of liquid-liquid phase separation (LLPS) (Sabari BR, et al. *Science*. 2018 361(6400):eaar3958.), and the condensate formation is nonlinear. The reduction in loops over long distances may reflect a reduced likelihood of condensates forming of SEs. We also observed that 68% of super-enhancers have reduced loop numbers in *Satb1* deficient cells (Fig. 4e) and about 80% of super-enhancers had reduced loop densities (average loop contacts) (Fig. S5g). So we think that SATB1 may interact with SEs and increase a higher chance of forming SE condensates.

b. How do the loop occurrence and loop intensity measurements come out of data like these? Fig. 4 and Fig. S5 show plots for loop number and loop length, but which data show the “loop intensity” measurement? How is the statistical confidence evaluated?

Response:

Each loop has a contact value, which generally is from paired-end tags (PETs). The loop intensity is the average contact of the SE-associated loops. Pearson correlation coefficient was used for calculating correlation. We added the information in figure legends. We added details in the results.

“To identify functional interactions among regulatory elements and their target promoters, we explored the chromatin loop structure using the program Fit-Hi-C(V2), a tool for assigning statistical confidence estimates to intra-chromosomal contact maps⁴³. We obtained 241088 and 103887 loop-like contacts from two WT Hi-C data, 289023 and 144941 contacts from two Satb1cKO Hi-C data. Pile-up analysis showed that the average strength of all chromatin loops displayed a mild reduction in Satb1-deficient DP thymocytes (Fig. 4c and S5e). The reduction was also observed in the loops associated promoters, super-enhancers, and Satb1 super-clusters. We noticed that loop sizes were reduced in Satb1 deficient cells with a median loop size of ~120 kb in WT and ~100 kb in Satb1cKO cells, which was also observed in E-P loops and SE-associated loops (Fig. 4d). Super-enhancers are clusters of enhancers occupied by a high density of transcription factors, co-factors, chromatin regulators, and RNA polymerase II⁴⁴. According to the phase separation model of super-enhancers, SEs may exist in two states in nuclei, one is a highly active condensate, and the other is a random loose state⁴⁴. Reduced sizes of SE-associated loops may be caused by more loose states of super-enhancers in nuclei after Satb1 deletion, which keeps short-distance chromatin interactions but loses long-distance interactions. It suggested that Satb1 may be involved in stabilizing interactions between regulatory elements and compacting chromatin condensates.”

c. Fig. 4c & f and Fig. S5d need much more explanation. These data are supposed to show the “Satb1 deletion increased loop numbers but reduced the loop strength” (lines 171-172), but this is not at all obvious. What are the axes? What are the numbers on the plots? Nothing is explained in the legends to these figures.

Response:

Thanks. It is an wrong description. We deleted it in the new version. More information was added in the new version.

“To identify functional interactions among regulatory elements and their target promoters, we explored the chromatin loop structure using the program Fit-Hi-C(V2), a tool for assigning statistical confidence estimates to intra-chromosomal contact maps⁴³. We obtained 241088 and 103887 loop-like contacts from two WT Hi-C data, 289023 and 144941 contacts from two Satb1cKO Hi-C data. Pile-up analysis showed that the average strength of all chromatin loops displayed a mile reduction in Satb1-deficient DP thymocytes (Fig. 4c and S5e). The reduction was also observed in the loops associated promoters, super-enhancers, and Satb1 super-clusters.

We noticed that loop sizes were reduced in Satb1 deficient cells with a median loop size of ~120 kb in WT and ~100 kb in Satb1cKO cells, which was also observed in E-P loops and SE-associated loops (Fig. 4d). Super-enhancers are clusters of enhancers occupied by a high density of transcription factors, co-factors, chromatin regulators, and RNA polymerase II⁴⁴. According to the phase separation model of super-enhancers, SEs may exist in two states in nuclei, one is a highly active condensate, and the other is a random loose state⁴⁴. Reduced sizes of SE-associated loops may be caused by more loose states of super-enhancers in nuclei after Satb1 deletion, which keeps close chromatin interactions but loses distal interactions. It suggested that Satb1 may be involved in stabilizing interactions between regulatory elements and compacting chromatin condensates.” in Page 8.

“c) Aggregate interactions and d) Boxplot of loop sizes of all loops, enhancer-promoter (E-P) loops, and super-enhancer (SE) loops identified from Hi-C data of Satb1WT thymocytes. Pile-up plots showing contacts of \pm 250kb regions around loops in WT or Satb1cKO Hi-C (10 kb resolution). The numbers in pile-up plots represent the value of the highest contacts in the heatmap. SE loops, loops with one or two anchors in WT-SE regions. **** p value < 0.0001 by two side Student’s t test.” in figure legends

d. Similarly, what are the dots shown in the scatter plots of Fig. 4e? Is each one an SE, and how are “loop numbers” counted for each SE? Are they raw numbers, or normalized somehow? This is not well explained either. Also, with these small changes, how great is the difference between WT and KO as compared to the difference between the two WT replicate samples, or the difference between the two KO samples?

Response:

Chromatin loops are statistically confident chromatin contacts ($p \leq 0.05$ and contact frequencies ≥ 5). The SE-loops are generated from normalized Hi-C data. Generally each SE has at least one loop. So we analyzed the loop number changes of each SE between WT and SATB1cKO. The plot here is to show that loop number and loop intensity are positively correlated with chromatin activity (H3K27ac) and gene expression. Of course, we can see that the number of loops in most SEs is reduced in SATB1deleted cells, but there are also many increases in the loop number and intensity of SEs. The loops used in Fig. 4e were generated from two sets of Hi-C data.

“c) Aggregate interactions and d) Boxplot of loop sizes of all loops, enhancer-promoter (E-P) loops, and super-enhancer (SE) loops identified from Hi-C data of Satb1WT thymocytes. Pile-up plots showing contacts of \pm 250kb regions around loops in WT or Satb1cKO Hi-C (10 kb resolution). The numbers in pile-up plots represent the value of the highest contacts in the heatmap. SE loops, loops with one or two anchors in WT-SE regions. **** p value < 0.0001 by two side Student’s t test.” in figure legends

“Pearson’s correlation analysis of super-enhancer associated H3K27ac signals (left) or relative gene expression (right) with numbers of loops associated with WT super-enhancers.” in figure legends

5. Typographical errors:

a. In both Fig. 5f and Fig. S6d, the labels for Bcl6 and Ets2 data are reversed. They are both high in DP cells, but Bcl6 is low in DN3 cells, Ets2 is increasing in DN3 cells.

b. Please label Fig. 6d to show WT and KO.

c. In line 228, please insert “6” to read “the Bcl6-SE”

Response:

Thanks. All are fixed in the new version.

6. In the Discussion, line 254, the authors assert that SATB1 is a “master regulator of DP thymocytes”. Indeed, their data show a role for SATB1 in the DP cells where its activity is highest, but it is very dramatic how many DP cells are produced even without SATB1, how well their transcriptomes co-cluster with WT DP cells, and how many of them manage to go on to make SP cells despite the complete lack of SATB1 in these lineages. This seems completely inconsistent with the normal implication of a “master regulator”. Of course this factor is worth studying, and it probably has great basic and clinical importance. But there is a huge difference between the phenotype when a paradigm-setting “master regulator” like EBF1 or PAX5 is knocked out in B cells, vs. this very subtle, modulating phenotype when SATB1 is knocked out in the T lineage.

Response:

SATB1 is indeed different from traditional master regulator like EBF1 or PAX5 in B cells. SATB1 regulates multiple processes in DP cells, including proliferation, *Tcra* rearrangement, Treg cell differentiation, positive and negative selection, etc. But it is not decisive for these processes. The role of SATB1 in DP thymocytes is versatile, similar to other chromatin organizing proteins such as CTCF. We revised this view in the Discussion.

“However, most of the factors only participate in one of biological processes. SATB1 regulates *Tcra* rearrangement⁶, positive and negative selection²⁷, and lineage decision in DP cells²⁵, which makes it as a versatile regulator of DP thymocytes.”

7. The authors elegantly showed that they can completely eliminate *Bcl6* and *Ets2* expression in DP cells by deletion of large SE-involved regions. But the statement that *Bcl6* and *Ets2* at high levels “play an essential role in DP cells” (line 267) similarly seems to be in contradiction with the very mild reduction in cell number and almost completely normal CD4/CD8 profiles of DP cells that have completely lost expression of these factors in Figs. 6c,d and 7b,c.

Response:

Thanks. The current data indeed do not support an essential role for these two genes and their SE in DP cells. We revised this view in the Discussion.

“Super-enhancer knockout mice confirmed that the high expressions of *Bcl6* and *Ets2* in DP cells play a role in DP cells. *Ets1* and *Ets2* are members of the *ets* family of transcription factors and play a role in thymocyte development, especial for DN-to-DP transition. *Ets1*^{-/-} mice displayed impaired DN3-to-DP transition due to susceptible to cell death of ND4⁶⁵.”

8. Finally, this is an optional point, but I wonder if the authors have considered the chance that the SATB1 KO phenotype could be relatively weak because of another factor in the same family that may be compensating for it when it is knocked out. Are there other factors with related roles that are also expressed in these cells? Even though SATB2 is not expressed there, I wonder if some of the other CUT-domain factors could be playing a supporting role.

Response:

It is a very interesting idea. There are indeed other CUT proteins that play important functions in the thymus, especially in DP cells. CCAAT displacement protein (CDP or CUX1) is a protein with three CUT domains, which is slightly different from SATB1 containing two CUT domains. CDP preferentially recognizes AT-rich DNA sequences that are often associated with matrix associating region (MAR) regions, similar to SATB1. In fact, it has been reported that both CUX1 and SATB1 bind a MAR region upstream of the TCR β enhancer, E β (Chattopadhyay S, Whitehurst CE, Chen J. A. J Biol Chem. 1998 ;273(45):29838-46.). CUX1 plays an essential role in thymocyte development and *Cux1* knockout mice had dramatically reduced thymic cellularity, with a preferential loss of CD4+CD8+ thymocytes (Sinclair, AM. et al. Blood (2001) 98 (13): 3658–3667.). CUX1 is a

transcription repressor and it was reported that TCR β MARbeta elements recruited Cux1 SMAR1 to repress E β -mediated recombination and transcription at the double positive stage of T cell development (Kaul-Ghanekar, R. et al. Nucleic Acids Res. (2004) ;32(16):4862-75.). It is possible that CUX1 supports the repression function of SATB1.

In summary, this paper is filled with valuable data, but the presentation right now is somewhat logically choppy, like three or four studies put together. It is too bad that in the current version, the hardest point to appreciate is the one that is stated in the title. It is hoped that the authors can clarify this.

Response:

We greatly appreciate these critical comments. These suggestions helped to improve the manuscript. We have provided new data and analysis in the new version and revised some representations, especially with regard to chromatin interactions. We believe that these modifications better support our conclusions.

REVIEWERS' COMMENTS

Reviewer #1 (Remarks to the Author):

The authors have provided new data and and revised a few representations, especially pertaining to chromatin interactions. They have satisfactorily addressed the reviewer's concerns.

Reviewer #2 (Remarks to the Author):

All my comments except major concern 1 (as Reviewer #2), which is necessary for broad headship and general interest in this field, have been appropriately addressed.

Reviewer #3 (Remarks to the Author):

The authors have painstakingly responded to a long list of requests by the reviewers, and they should be thanked for their patient and thoughtful replies. Their work is very valuable, providing a rich resource of information about the action of a key transcription factor, SATB1, that has still retained some mysteries. This paper brings current understanding of SATB1's roles up to date, with global assessments of its linkages to superenhancers and chromatin looping features in DP thymocyte chromatin as well as diagnostic histone marks for particular modes of regulation. The data included for SATB1's regulation of two other transcription factor genes, Ets2 and Bcl6, is also a key contribution. The authors have worked hard to improve the manuscript to make their findings clearer. Some final suggestions follow.

1. Reviewing this set of reviewer responses, it seems that certain basic points that may be obvious to the authors were not evident to some readers. To make sure that the logic flow of the paper is understood better by future readers, it would be helpful to make these points explicitly.

a. For clarity, the authors could spell out one thing out at the start. Because TCRA recombination was already known to be affected by SATB1, any gene expression that depended on TCRA-mediated positive selection (DP stage to SP stage transition) would also be expected to be altered, even if the effect was only indirect. That might or might not necessarily say much about SATB1's direct activity as a transcription factor. However, the main interesting point here is how greatly SATB1 deficiency affects

the DP state itself, preceding positive selection. This is important, and it cannot only be due to effects on Rag and TCRA. Therefore, the emphasis in this paper is on how SATB1 works across the genome to regulate a broad range of DP-stage-associated genes.

b. One other point that would be useful to spell out at the start is that SATB1 is known to be implicated biochemically in gene repression as well as in gene activation. Therefore, its direct targets are not expected to show homogeneous chromatin changes. Instead, directly regulated targets would be expected to fall into at least two classes, some showing changes associated with activation and others showing changes associated with repression.

2. The authors are thanked for adding much better figure legends and supplementary figure legends. However, there are still no legends for the Supplementary Datasets. It is not even clear how to refer to these tables because many of them are not labeled. The first, second, and sixth datasets uploaded have no identification as to which datasets they are, and the third dataset is called Supplementary Dataset 2. The contents of these tables are extremely important to clarify. The lack of legends defining the column entries makes the data in tables like the first (nameless) Supplementary Dataset almost impossible to use. Also, the genes labeled with “DN3” and “DN1” categories in the second (nameless) dataset are not explained at all. Are these the full set of signature genes for these cell types, or are they the genes that overlap with Satb1 KO-upregulated genes?

3. In the vitally important first supplementary dataset, the one showing the genes that are calculated to be differentially expressed between cluster 3 and cluster 0 in the scRNA-seq analysis, there is no key to what the values in each of the columns represent. They need to be explained. Some results are also very surprising, and it is not clear whether these are misprints or due to the particular ways that column values were calculated. This needs to be reported.

a. Throughout the first sheet of this dataset, the numerical value for each gene (fraction of cells expressing??) is almost always substantially less in cluster 3 than the value for that gene in cluster 0. Even so, the signs of the “avg_log2FC values” for those genes go back and forth between positive and negative from one gene to the next, suggesting opposite patterns of abundance. How can this be? What does this “avg_log2FC” measure?

b. Why are Bcl6, Ets2, and Ccr9 not showing up in this first dataset? Fig. S1 clearly shows these genes to be differentially expressed, with much denser expression in the cluster 0 region than in the cluster 3 region.

c. It is very important to explain what parameters were used in the pipeline to generate these Tables, for example the pseudocount and minimum positive percentage thresholds for the scRNA-seq results, etc.

4. In Fig. 1f, the axis label is confusing. If the KO results in upregulation of DN1 and DN3 genes, then why are the results shown with the highest value associated with WT, to the left, and a strongly negative enrichment score for the KO on the right? A negative enrichment score would suggest that these genes were preferentially downregulated in the KO. Were these axes reversed?

5. With regard to Fig. S3f, why not simply say that it is possible that genes linked to cluster 4 sites could be direct SATB1 repression targets? This would certainly be a clarifying point to make if there is there any other evidence for this response by the genes linked to cluster 4 sites. It is far easier to explain why this group of sites show opposite modifications from the majority if they actually represent the sites with an opposite function.

Overall, I think that this paper contributes a great deal of valuable information. I am a little sorry that the unique features of the cluster 3 cells in Fig. 1a were not identified and explained further, and that the mechanisms of SATB1's repressive effects were not investigated directly. But this paper is filled with valuable results and thoughtful analyses, and it should be of strong interest to many readers.

REVIEWERS' COMMENTS

Reviewer #1 (Remarks to the Author):

The authors have provided new data and revised a few representations, especially pertaining to chromatin interactions. They have satisfactorily addressed the reviewer's concerns.

Response:

We appreciate the reviewers for their critical comments, which greatly improved the quality of our article.

Reviewer #2 (Remarks to the Author):

All my comments except major concern 1 (as Reviewer #2), which is necessary for broad headship and general interest in this field, have been appropriately addressed.

Response:

We appreciate the reviewers for their comments, which greatly improved the quality of our article.

Reviewer #3 (Remarks to the Author):

The authors have painstakingly responded to a long list of requests by the reviewers, and they should be thanked for their patient and thoughtful replies. Their work is very valuable, providing a rich resource of information about the action of a key transcription factor, SATB1, that has still retained some mysteries. This paper brings current understanding of SATB1's roles up to date, with global assessments of its linkages to superenhancers and chromatin looping features in DP thymocyte chromatin as well as diagnostic histone marks for particular modes of regulation. The data included for SATB1's regulation of two other transcription factor genes, Ets2 and Bcl6, is also a key contribution. The authors have worked hard to improve the manuscript to make their findings clearer. Some final suggestions follow.

Response:

We appreciate the reviewers for their comments, which greatly improved the quality of our article.

1. Reviewing this set of reviewer responses, it seems that certain basic points that may be obvious to the authors were not evident to some readers. To make sure that the logic flow of the paper is understood better by future readers, it would be helpful to make these points explicitly.

a. For clarity, the authors could spell out one thing out at the start. Because TCRA recombination was already known to be affected by SATB1, any gene expression that depended on TCRA-mediated positive selection (DP stage to SP stage transition) would also be expected to be altered, even if the effect was only indirect. That might or might not necessarily say much about SATB1's direct activity as a transcription factor. However, the main interesting point here is how greatly SATB1 deficiency affects the DP state itself, preceding positive selection. This is important, and it cannot only be due to effects on Rag and TCRA. Therefore, the emphasis in this paper is on how SATB1 works across the genome to regulate a broad range of DP-stage-associated genes.

Response:

It is a nice suggestion.

“Although *Tcra* rearrangement deficiency affects $\alpha\beta$ TCR-mediated selection and thus affects many gene expressions indirectly, STAB1 deficiency should not affect DP cells only through RAG expression and *Tcra* rearrangement.” was added in page 4 line 9.

b. One other point that would be useful to spell out at the start is that SATB1 is known to be implicated biochemically in gene repression as well as in gene activation. Therefore, its direct targets are not expected to show homogeneous chromatin changes. Instead, directly regulated targets would be expected to fall into at least two classes, some showing changes associated with activation and others showing changes associated with repression.

Response:

It is a nice suggestion. It is in the introduction part.

“SATB1 can activate or repress gene transcription by recruiting p300/CBP-associated factor (PCAF) or histone deacetylase, respectively²⁹.”

2. The authors are thanked for adding much better figure legends and supplementary figure legends. However, there are still no legends for the Supplementary Datasets. It is not even clear how to refer to these tables because many of them are not labeled. The first, second, and sixth datasets uploaded have no identification as to which datasets they are, and the third dataset is called Supplementary Dataset 2. The contents of these tables are extremely important to clarify. The lack of legends defining the column entries makes the data in tables like the first (nameless) Supplementary Dataset almost impossible to use. Also, the genes labeled with “DN3” and “DN1” categories in the second (nameless) dataset are not explained at all.. Are these the full set of signature genes for these cell types, or are they the genes that overlap with *Satb1* KO-upregulated genes?

Response:

It is a nice suggestion. We fixed the error and add a doc file named "Description of Additional Supplementary Files" for supplementary data.

3. In the vitally important first supplementary dataset, the one showing the genes that are calculated to be differentially expressed between cluster 3 and cluster 0 in the scRNA-seq analysis, there is no key to what the values in each of the columns represent. They need to be explained. Some results are also very surprising, and it is not clear whether these are misprints or due to the particular ways that column values were calculated. This needs to be reported.

a. Throughout the first sheet of this dataset, the numerical value for each gene (fraction of cells expressing??) is almost always substantially less in cluster 3 than the value for that gene in cluster 0. Even so, the signs of the "avg_log2FC values" for those genes go back and forth between positive and negative from one gene to the next, suggesting opposite patterns of abundance. How can this be? What does this "avg_log2FC" measure?

Response:

It is an interesting question. There are two indicators characterizing differential gene expression in scRNA-seq data. One is avg_log2FC, log fold-change of the average expression between the two groups. We use log fold-change of the average expression for upregulation or downregulation of genes. The other indicator is pct, the percentage of cells where the feature is detected. Due to the depth of sequencing, not all expressed genes can be detected in every cell. Genes that are highly expressed have a higher chance of being detected in cells. However, pct is also affected by cell size. Larger cells contain more mRNA, and the probability of the gene detected in cells is higher. During the development of DP cells, the cell size has undergone changes. In the early stage, it is called large DP because of the large cell size. As the development progresses, the cells become smaller, which is called small DP. Due to the loss of SATB1, the development of some cells was blocked, and the proportion of large DP was higher overall. Cluster 3 represents larger cells, while cluster 0 represents smaller cells. So the pct of cluster 3 is higher than that of cluster 0. The labels of cluster3 and cluster 0 are reversed, which is fixed in the new version.

b. Why are Bcl6, Ets2, and Ccr9 not showing up in this first dataset? Fig. S1 clearly shows these genes to be differentially expressed, with much denser expression in the cluster 0 region than in the cluster 3 region.

Response:

It is a good question. We checked the average expression of Ets2, Bcl6 and Ccr9 in cluster 0 and cluster 3: Ets2, 2.18918487 in cluster 0, 0.41022197 in cluster 3; Bcl6, 0.50050576 in cluster 0, 0.12082239 in cluster 3; Ccr9, 14.32855695 in cluster 0, 15.12695896 in cluster 3. It is true that Bcl6 and Ets2 are downregulated in cluster 3. The reason why Bcl6 and Ets2 were not in the list of differentially expressed genes is the limitation of Seurat in single-cell RNA-seq analysis. Due to computational constraints, Seurat could only select 2000 genes for analysis. Because of some

conditional constraints, Bcl6 and Ets2 were not selected into the 2000 genes, so the differentially expressed genes did not contain these two genes. So the list is marker genes instead of differentially expressed genes.

c. It is very important to explain what parameters were used in the pipeline to generate these Tables, for example the pseudocount and minimum positive percentage thresholds for the scRNA-seq results, etc.

Response:

Thanks for the suggestion. The conditions are added.

“FindMarkers function in Seurat was used for identification of differentially expressed genes with default conditions (logfc.threshold = 0.25, test.use = "wilcox", min.pct = 0.1, min.diff.pct = -Inf, verbose = FALSE, only.pos = FALSE, max.cells.per.ident = Inf, random.seed = 1, latent.vars = NULL, min.cells.feature = 3, min.cells.group = 3, pseudocount.use = 1).”

4. In Fig. 1f, the axis label is confusing. If the KO results in upregulation of DN1 and DN3 genes, then why are the results shown with the highest value associated with WT, to the left, and a strongly negative enrichment score for the KO on the right? A negative enrichment score would suggest that these genes were preferentially downregulated in the KO. Were these axes reversed?

Response:

Thanks. We changed the labels with “down-regulated genes in Satb1cKO” at the left and “up-regulated genes in Satb1cKo” at the right in Fig. 1f. The DN1 and DN3 genes enriched in up-regulated genes (right) are more than that in down-regulated genes (right).

5. With regard to Fig. S3f, why not simply say that it is possible that genes linked to cluster 4 sites could be direct SATB1 repression targets? This would certainly be a clarifying point to make if there is there any other evidence for this response by the genes linked to cluster 4 sites. It is far easier to explain why this group of sites show opposite modifications from the majority if they actually represent the sites with an opposite function.

Response:

Thanks. This is a good suggestion, but we cannot directly describe that they are direct SATB1 repression targets because we did not analyze the relationship between SATB1-repressed regions and expression changes of their associating genes. Since this study focused on the regulatory function of SATB1 on DP cell identity genes, those genes repressed by SATB1 were not further investigated and the corresponding data were insufficient to draw definitive conclusions.

Overall, I think that this paper contributes a great deal of valuable information. I am a little sorry that the unique features of the cluster 3 cells in Fig. 1a were not identified and explained further, and that the mechanisms of SATB1's repressive effects were not investigated directly. But this paper is filled with valuable results and thoughtful analyses, and it should be of strong interest to many readers.

Response:

Thanks. The feature of the cluster 3 and the mechanisms of SATB1's repressive function are interesting questions and we will investigate it later.